# STATISTICAL TRACTABILITY OF OFF-POLICY EVALUATION OF HISTORY-DEPENDENT POLICIES IN POMDPS

**Yuheng Zhang & Nan Jiang**
University of Illinois Urbana-Champaign
{yuhengz2,nanjiang}@cs.illinois.edu

## ABSTRACT

We investigate off-policy evaluation (OPE), a central and fundamental problem in reinforcement learning (RL), in the challenging setting of Partially Observable Markov Decision Processes (POMDPs) with large observation spaces. Recent works of Uehara et al. (2023a); Zhang & Jiang (2024) developed a *model-free* framework and identified important coverage assumptions (called *belief* and *outcome* coverage) that enable accurate OPE of *memoryless* policies with polynomial sample complexities, but handling more general target policies that depend on the entire observable history remained an open problem. In this work, we prove information-theoretic hardness for model-free OPE of history-dependent policies in several settings, characterized by additional assumptions imposed on the behavior policy (memoryless vs. history-dependent) and/or the state-revealing property of the POMDP (single-step vs. multi-step revealing). We further show that some hardness can be circumvented by a natural *model-based* algorithm—whose analysis has surprisingly eluded the literature despite the algorithm's simplicity—demonstrating provable separation between model-free and model-based OPE in POMDPs.

## 1 INTRODUCTION

Off-policy evaluation (OPE) aims to evaluate a target policy $\pi_e$ using an offline dataset collected by a different behavior policy $\pi_b$. The problem plays a crucial role in reinforcement learning (RL), and is particularly relevant to real-world scenarios where policies need to be properly evaluated before online deployment (Murphy, 2003; Ernst et al., 2006; Mandel et al., 2014; Bottou et al., 2013; Chapelle et al., 2014; Theocharous et al., 2015).

Efficient OPE requires the behavior policy $\pi_b$ to satisfy certain coverage assumptions with respect to the target policy $\pi_e$. In the setting of Markov Decision Processes (MDPs), it is well established that a bounded state-action density ratio between $\pi_e$ and $\pi_b$ suffices for polynomial sample-complexity bounds; see Uehara et al. (2022b); Jiang & Xie (2024) for surveys and tutorials on the topic. However, the Markov assumption, that the immediate observation is a sufficient statistic of history, can be restrictive in scenarios where the state is latent and unobservable to the agent, as is often the case in many real-world applications.

In this paper, we study OPE in non-Markov environments modelled as partially observable MDPs (POMDPs),[1] where the observation space is large and demands the use of function approximation. In POMDPs, the agent only has access to observations rather than the latent state, and the next observation may depend on the entire history of observation-action sequences (or simply, the history). A common approach to apply MDP techniques is to treat the history as the state, thereby reducing a POMDP to a history-based MDP. However, under this conversion, the state-action density ratio becomes the density ratio of the entire observation-action sequence, which grows exponentially with the horizon length.

---

[1]When we refer to OPE in POMDPs, we mean *unconfounded* POMDPs, where the behavior policy only depends on the observable variables and not the latent state. There is also research on OPE in confounded POMDPs, where the behavior policy $\pi_b$ depends on (*only*) the latent state (Shi et al., 2022; Bennett & Kallus, 2024); see Zhang & Jiang (2024) for further discussions on the distinction between the two settings.

Table 1: Summary of whether $\mathrm{poly}(H, \log(|\mathcal{M}|/\delta), \epsilon, C_{\mathcal{A}}, C_{\square}, C_{\mathcal{H}})$ (c.f. Theorem 1) complexity is achievable in different settings, where $C_{\square}$ is either $C_{\mathcal{O}}$ or $C_{\mathcal{F}}$ depending on whether single-step or multi-step revealing is assumed. "MF" and "MB" stand for model-free (Definition 2) and model-based (Section 4), respectively. "✓" indicates positive results, and "✗" indicates information-theoretic hardness. The setting is the easiest in the top-left corner, and becomes harder in the right or down direction. Therefore, the hardness of MF in Row 2 automatically implies those in Row 3. The bottom-right corner for MB is an open problem which we conjecture to be intractable.

| Policy Types | Single-step Revealing | Multi-step Revealing |
|---|---|---|
| Memoryless $\pi_b$ & $\pi_e$ | MF: ✓ (Zhang & Jiang, 2024), MB: ✓ (Theorem 5) | |
| Memoryless $\pi_b$ & History-dependent $\pi_e$ | MF: ✗ (Theorem 3), MB: ✓ (Theorem 5) | |
| History-dependent $\pi_b$ & $\pi_e$ | MF: ✗ , MB: ✓ (Theorem 4) | MF: ✗ , MB: ? |

To address this issue, a recent line of research (Uehara et al., 2023a; Zhang & Jiang, 2024) has proposed model-free methods for OPE in POMDPs with large observation spaces. Zhang & Jiang (2024) identify two novel coverage assumptions for OPE in POMDPs, belief and outcome coverage, and demonstrate that their algorithm achieves polynomial sample complexity under these assumptions. However, they focus on evaluating *memoryless* target policies $\pi_e$, which ignore history and depend only on the current observation. Extension to history-dependent $\pi_e$ exists, but the guarantees quickly deteriorate when the history window that $\pi_e$ depends on has a nontrivial length (Uehara et al., 2023a, Appendix C). This motivates us to study the following question:

*When can we achieve polynomial sample complexity for OPE of history-dependent target policies?*

We investigate the question in a range of concrete settings, and the answer turns out to be more complex than a simple yes and no. These settings are defined by variations along several dimensions:

- **Model-free vs. model-based algorithms**  The algorithms in Uehara et al. (2023a); Zhang & Jiang (2024) are *model-free*, in the sense that the algorithm only queries $\pi_e$ on histories in the offline dataset. Under this rather broad definition (Definition 3 in Zhang & Jiang (2024)), we show that ***no model-free algorithms can handle general history-dependent target policies***, even if we impose additional assumptions to make the problem easier in other dimensions (see below). This motivates us to also consider model-based algorithms that fit a POMDP model from data, which circumvent the hardness as they query $\pi_e$ on model-generated synthetic trajectories.

- **Single-step vs. multi-step (outcome) revealing** The outcome coverage condition identified by Zhang & Jiang (2024) asserts that the future observation-action sequences can probabilistically decode the latent state (Assumption 9 in Zhang & Jiang (2024)). A stronger version of the condition is that the immediate observation suffices, which corresponds to a standard (single-step) "revealing" assumption commonly made in online RL for POMDPs (Liu et al., 2022a).

- **Memoryless vs. history-dependent $\pi_b$**  Another dimension is whether $\pi_b$ is also history dependent. In the literature, it has been reported that a history-dependent $\pi_b$ often makes it difficult to infer POMDP dynamics from data, and a memoryless $\pi_b$ makes it easier to do so (Kwon et al., 2024).

Our findings are summarized in Table 1. With either relaxation (single-step revealing *or* memoryless $\pi_b$), a simple Maximum Likelihood Estimation (MLE)-based model-based algorithm achieves desired guarantees, where all model-free algorithms must suffer hardness. To our best knowledge, these results are the first polynomial sample-complexity bound for evaluating history-dependent target policies under coverage assumptions, and demonstrate a formal separation between model-based and model-free OPE in POMDPs.

## 2 PRELIMINARIES

**Notation.**    For a vector $\mathbf{a}$, we use $\mathrm{diag}(\mathbf{a})$ to denote the diagonal matrix with $\mathbf{a}$ as the diagonal and use $[\mathbf{a}]_i$ to denote its $i$-th element. We use $\mathbf{e}_i$ to denote the basis vector with the $i$-th element being one. For a positive integer $n$, we use $[n]$ to denote the set $\{1, 2, \cdots, n\}$. For a matrix $M$, we use $\sigma_{\min}(M)$ and $M^\dagger$ to denote its minimum singular value and pseudo-inverse respectively. The $ij$

entry of matrix $M$ is denoted as $[M]_{ij}$ and the $i$-th row of $M$ is denoted as $[M]_{i,:}$. The $L_1$ norm of matrix $M$ is $\|M\|_1 = \sup_{\mathbf{x} \neq \mathbf{0}} \frac{\|M\mathbf{x}\|_1}{\|\mathbf{x}\|_1}$.

**POMDP Setup.** We use tuple $\left\langle H, \mathcal{S} = \bigcup_{h=1}^H \mathcal{S}_h, \mathcal{A}, \mathcal{O} = \bigcup_{h=1}^H \mathcal{O}_h, R, \mathbb{O}, \mathbb{T}, d_1 \right\rangle$ to specify a finite-horizon POMDP. Here $H$ is the length of horizon; $\mathcal{S}_h$ is the state space at step $h$; $\mathcal{A}$ is the action space with $|\mathcal{A}| = A$; $\mathcal{O}_h$ is the observation space at step $h$ with $|\mathcal{O}_h| = O$; $R : \mathcal{O} \to [0, 1]$ is the reward function; $\mathbb{T} = \{\mathbb{T}_h\}_{h \in [H-1]}$ is the collection of transition dynamics where $\mathbb{T}_h : \mathcal{S}_h \times \mathcal{A} \to \mathcal{S}_{h+1}$; $\mathbb{O} = \{\mathbb{O}_h\}_{h \in [H]}$ is the collection of emission dynamics where $\mathbb{O}_h : \mathcal{S}_h \to \mathcal{O}_h$; $d_1 \in \Delta(\mathcal{S}_1)$ is the distribution of initial state $s_1$. All state, action, and observation spaces are finite and discrete. However, since the observation space can be very rich, the cardinality $O$ may be arbitrarily large. Therefore, we aim to obtain sample complexity bounds that avoid any explicit dependence on $O$.

At the beginning of each episode in the POMDP, an initial state $s_1$ drawn from $d_1$. At each step $h$, the decision-making agent observes $o_h \sim \mathbb{O}_h(\cdot \mid s_h)$, along with the reward $r_h(o_h)$, and then takes an action $a_h$. After this, the environment transitions to the next state $s_{h+1} \sim \mathbb{T}_h(\cdot \mid s_h, a_h)$, and the episode terminates after $a_H$. Note that the states $s_{1:H}$ are latent and unobservable to the agent.

**History and Belief State.** We use $\tau_h = (o_1, a_1, \cdots, o_h, a_h) \in \mathcal{T}_h := \prod_{h'=1}^h (\mathcal{O}_h \times \mathcal{A})$ to denote the historical observation-action sequence up to step $h$. Given history $\tau_h$, we define $\mathbf{b}_\mathcal{S}(\tau_h) \in \mathbb{R}^{|\mathcal{S}_{h+1}|}$ as its belief state vector and $[\mathbf{b}_\mathcal{S}(\tau_h)]_i = \mathbb{P}(s_{h+1} = i \mid \tau_h)$. [2]

**Policies.** A (history-dependent) policy $\pi = \{\pi_h\}_{h \in [H]}$ where $\pi_h : \mathcal{T}_{h-1} \times \mathcal{O}_h \to \Delta(\mathcal{A})$ specifies the action probability given the history $\tau_{h-1}$ and the current observation $o_h$. A *memoryless* policy only depends on the current observation (i.e., $\pi_h : \mathcal{O}_h \to \Delta(\mathcal{A})$). We define $J(\pi)$ as the expected cumulative return under policy $\pi$: $J(\pi) := \mathbb{E}_\pi \left[ \sum_{h=1}^H R(o_h) \right]$. Here we use $\mathbb{E}_\pi$ to denote the expectations under policy $\pi$, $\mathbb{P}^\pi(\cdot)$ for the probability of an event under the same policy and $d_h^\pi(\cdot)$ for the marginal distribution of $s_h, a_h$ under $\pi$.

**The Outcome Matrix.** For any step $h \in [H]$, we use $f_h = (o_h, a_h, \cdots, o_{H-1}, a_{H-1}, o_H) \in \mathcal{F}_h := \prod_{h'=h}^{H-1} (\mathcal{O}_{h'} \times \mathcal{A}) \times \mathcal{O}_H$ to denote the future after step $h$. For future $f_h$, we define $\mathbf{u}(f_h) \in \mathbb{R}^{|\mathcal{S}_h|}$ as its outcome vector where $[\mathbf{u}(f_h)]_i = \mathbb{P}^{\pi_b}(f_h \mid s_h = i)$. Then we define the outcome matrix $U_{\mathcal{F},h} \in \mathbb{R}^{|\mathcal{F}_h| \times |\mathcal{S}_h|}$ where the row indexed by $f_h$ is $\mathbf{u}(f_h)$. Note that the outcome matrix $U_{\mathcal{F},h}$ depends on the behavior policy $\pi_b$, which we omit in the notation.

**Off-policy Evaluation (OPE).** In OPE, we aim to use an offline dataset collected by a behavior policy $\pi_b$ to estimate the expected cumulative return of a target policy $\pi_e$, which is $J(\pi_e)$. The dataset $\mathcal{D}$ consists of $n$ data trajectories $\{(o_1^{(i)}, a_1^{(i)}, r_1^{(i)}, \ldots, o_H^{(i)}, a_H^{(i)}, r_H^{(i)}) : i \in [n]\}$. Without loss of generality, we assume the reward function $R$ is known and we need to learn the emission and transition dynamics. In this paper, we focus on evaluating history-dependent $\pi_e$, which is a more general setting compared to the memoryless $\pi_e$ considered in previous works (Uehara et al., 2023a; Zhang & Jiang, 2024). Therefore, the action $a_h$ depends on both the history $\tau_{h-1}$ and the observation $o_h$. Throughout the paper, we make the following assumption:

**Assumption A.** We assume $\pi_b(a_h \mid \tau_{h-1}, o_h)$ is known and $\max_{h, \tau_{h-1}, o_h, a_h} \frac{1}{\pi_b(a_h \mid \tau_{h-1}, o_h)} \leq C_\mathcal{A}$.

The parameter $C_\mathcal{A}$ will not always show up in our upper bounds due to its looseness,[3] but it is a useful relaxation for making meaningful comparisons across different settings of interest in Table 1.

**Learning Goal.** We consider rich and large observation spaces and want to avoid paying explicit dependence on $O$ in the sample complexity. One way of achieving so is to use importance sampling

---

[2]The belief state is determined by the environment dynamics and is independent of the policy. Our definition is the same as in Uehara et al. (2023a); Zhang & Jiang (2024); this is slightly different from the usual definition that also includes $o_{h+1}$ as a conditional variable.

[3]See discussion of uniform vs. policy-specific coverage in Section 3.1; the policy-specific counterpart of Assumption A is to bound $\pi_e(a_h \mid \tau_{h-1}, o_h)/\pi_b(a_h \mid \tau_{h-1}, o_h)$.

(Precup, 2000), which avoids $O$ but pays an exponential-in-horizon quantity $O(C_{\mathcal{A}}{}^{H})$ instead. We follow the setting of Uehara et al. (2023a); Zhang & Jiang (2024) and use function approximation to avoid both $O$ and exponential-in-$H$, by assuming that the learner has access to a realizable model class $\mathcal{M}$.[4] More formally, we make the following assumption throughout (except in Section 5):

**Assumption B** (Realizability). We assume that the learner is given $\mathcal{M}$, a class of POMDPs with the same $\mathcal{S}, \mathcal{A}, \mathcal{O}, H, R$ components as $M^{\star}$. Furthermore, $M^{\star} \in \mathcal{M}$.

Our goal is to achieve sample complexity bounds polynomial in $\log|\mathcal{M}|, H$ as well as appropriate coverage parameters introduced in the next section.

## 3    INFORMATION-THEORETIC HARDNESS OF MODEL-FREE ALGORITHMS

We start by introducing the key assumptions that enable efficient OPE of *memoryless* policies in prior works. Then, in Section 3.2, we demonstrate via a lower bound that evaluating history-dependent $\pi_e$ is information-theoretically hard for *any* model-free algorithm (Theorem 3), motivating the investigation of model-based algorithms in Section 4.

### 3.1    KEY ASSUMPTIONS AND EXISTING RESULTS FOR MEMORYLESS $\pi_e$

The prior work of Zhang & Jiang (2024) identify two key assumptions for OPE of memoryless policies in POMDPs, as introduced below.

**Assumption C** (Uniform Belief Coverage). Define

$$\Sigma_{\mathcal{H},h} = \mathbb{E}_{\pi_b}[\mathbf{b}_{\mathcal{S}}(\tau_{h-1})\mathbf{b}_{\mathcal{S}}(\tau_{h-1})^{\top}].$$

Assume that $1/\sigma_{\min}(\Sigma_{\mathcal{H},h}) \leq C_{\mathcal{H}}, \forall h \in [H-1]$ for some $C_{\mathcal{H}} < \infty$.

**Assumption D** (Multi-step Outcome Revealing). Define[5]

$$\Sigma_{\mathcal{F},h} := U_{\mathcal{F},h}^{\top} Z_h^{-1} U_{\mathcal{F},h}, \quad \text{where} \quad Z_h := \text{diag}(U_{\mathcal{F},h}\mathbf{1}_{\mathcal{S}_h}).$$

Here $\text{diag}(\cdot)$ is a diagonal matrix with its diagonal being the input vector, and $\mathbf{1}_{\mathcal{S}_h}$ is the all-1 vector with dimension $|\mathcal{S}_h|$. We assume that $\|\Sigma_{\mathcal{F},h}^{-1}\|_1 \leq C_{\mathcal{F}}, \ \forall h \in [H-1]$ for some $C_{\mathcal{F}} < \infty$.

Intuitively, Assumption C states that the belief vector $\mathbf{b}_{\mathcal{S}}(\tau_{h-1})$ spans all directions of $\mathbb{R}^{|\mathcal{S}_h|}$ when $\tau_{h-1}$ is generated with $\pi_b$, and similar "linear coverage" conditions are also found in the linear MDP literature (Jin et al., 2021; Xiong et al., 2022a). Assumption D is an analogous condition but stated for the future after step $h$, and can be interpreted as that the future $f_h$ can nontrivially predict the latent state $s_h$ (thus the term "revealing"). In fact, when $f_h$ can deterministically predict $s_h$, Assumption D holds with $\Sigma_{\mathcal{F},h} = \mathbf{I}$ (the identity matrix) and $C_{\mathcal{F}} = 1$.

**Uniform vs. Policy-specific Coverage**    Both assumptions above are only properties of the behavior policy $\pi_b$, whereas the original conditions of Zhang & Jiang (2024) are tighter versions that account for the properties of $\pi_e$ as well. For example, their belief coverage does not require covering all directions of $\mathbb{R}^{|\mathcal{S}_h|}$, but only the direction of $[d_h^{\pi_e}(s_h = s)]_{s \in \mathcal{S}_h}$. We call the former *uniform coverage* (since it works for all target policies) and the latter *policy-specific coverage* (since it is specific to the $\pi_e$ under consideration). Our positive results in Section 4 also depend on policy-specific coverage parameters, but they are different from the ones in Zhang & Jiang (2024) due to the technical differences between model-free and model-based analyses. Moreover, some definitions (e.g., outcome coverage in Zhang & Jiang (2024), which is the policy-specific version of our Assumption D) do not extend to history-dependent $\pi_e$. Therefore, we introduce the uniform version of the assumptions here for a clean and fair comparison.

---

[4]The algorithms in Uehara et al. (2023a); Zhang & Jiang (2024) are model-free and require value function and Bellman error classes; in Appendix A we show how these classes can be constructed automatically from $\mathcal{M}$, which is a common practice when comparing across model-based and model-free algorithms (Chen & Jiang, 2019; Sun et al., 2019).

[5]The definitions here are based on $U_{\mathcal{F},h}$, calculated using the dynamcis of $M^{\star}$. Later we will also be interested in variant of this assumption, where all quantities are replaced by their counterparts calculated using the dynamics of some candidate model $M \in \mathcal{M}$.

We now state the result of Zhang & Jiang (2024) in our setup (see Footnote 4 for how we invoke their model-free algorithms and analyses in our model-based setup), with proof in Appendix A.

**Theorem 1** (Corollary of Theorem 7 of Zhang & Jiang (2024)). *Under Assumptions C and D, assume that $\pi_b$ and $\pi_e$ are memoryless, there exists an algorithm (see Appendix A) such that, with probability at least $1 - \delta$,[6] $|J(\pi_e) - \widehat{J}(\pi_e)| \leq \epsilon$ with a sample complexity $n = \mathrm{poly}(H, \log(|\mathcal{M}|/\delta), \epsilon, C_{\mathcal{A}}, C_{\mathcal{F}}, C_{\mathcal{H}})$, where $\widehat{J}(\pi_e)$ is the algorithm's estimation of $J(\pi_e)$.*

### 3.2 HARDNESS RESULTS FOR HISTORY-DEPENDENT $\pi_e$

We now show that Theorem 1 cannot hold if we make $\pi_b$ and $\pi_e$ general history-dependent policies. In fact, the hardness is not specific to the algorithms of Uehara et al. (2023a); Zhang & Jiang (2024), but applies generally to a broad range of *model-free* algorithms, as defined below.

**Definition 2** (Model-free algorithm). We say that an algorithm for OPE in POMDPs is *model-free*, if it only queries the action distribution of $\pi_e$ on trajectories observed in the offline dataset, i.e., the only information known about $\pi_e$ is $\pi_e(\cdot|o_1^{(i)}, a_1^{(i)}, \ldots o_h^{(i)})$ for all $h \in [H], i \in [n]$.

This definition is satisfied by the algorithms of Uehara et al. (2023a); Zhang & Jiang (2024). When it is specialized to the MDP setting, it is also satisfied by most standard algorithms considered model-free, such as importance sampling (Precup, 2000), Fitted-Q Evaluation (Ernst et al., 2005; Le et al., 2019; Voloshin et al., 2019), and marginalized importance sampling Liu et al. (2018); Uehara et al. (2020). The spirit of focusing on how algorithms access and process information in the input is consistent with prior works that show model-free and model-based separation from a learning-theoretic perspective (Chen & Jiang, 2019; Sun et al., 2019).

Moreover, the hardness still holds even if we make the problem easier in two aspects:

1. We keep $\pi_b$ memoryless (the more general setting is that $\pi_b$ is also history-dependent just as $\pi_e$).
2. Assumption D states that (multi-step) future can reveal the latent state $s_h$. The online POMDP literature has a related and stronger assumption that the immediate observation $o_h$ plays the same role (Liu et al., 2022a):

**Assumption E** (Single-step Outcome Revealing). Define

$$\Sigma_{\mathcal{O},h} := \mathbb{O}_h^\top W_h^{-1} \mathbb{O}_h, \quad \text{where} \quad W_h := \mathrm{diag}(\mathbb{O}_h \mathbf{1}_{\mathcal{S}_h}).$$

We assume $\|\Sigma_{\mathcal{O},h}^{-1}\|_1 \leq C_{\mathcal{O}} \; \forall h \in [H-1]$ for some $C_{\mathcal{O}} < \infty$.

The interpretation of this assumption is similar to multi-step revealing Assumption D, but requires that the immediate observation $o_h$ (instead of the entire future as in multi-step revealing) is sufficient for making nontrivial predictions of $s_h$. Therefore, the single-step outcome revealing assumption (Assumption E) should be treated as generally stricter than its multi-step counterpart (Assumption D), though a rigorous and quantitative comparison between $C_{\mathcal{O}}$ and $C_{\mathcal{F}}$ is somewhat complicated and we defer the discussion to Appendix G. Variants of this assumption have been proposed in online learning in POMDPs (Liu et al., 2022a), but the original version has a poor scalability w.r.t. the number of observations, as pointed out by Chen et al. (2022) and Zhang & Jiang (2024); see Example 1 of Zhang & Jiang (2024) for further discussions. Our Assumption 4 fixes the issue by using a similar inverse-weighting scheme as Assumption 5 and enjoys better scaling with the size of the observation space. For example, when $o_h$ can uniquely determine $s_h$, $\Sigma_{\mathcal{O},h} = \mathbf{I}$ and is independent of the number of observations.

**Theorem 3** (Information-theoretic hardness of model-free algorithms). *In the same setup as Theorem 1, if we allow $\pi_e$ to be history-dependent (but $\pi_b$ is still memoryless), no model-free algorithm can achieve the polynomial sample complexity guarantee. This still holds even if we replace $C_{\mathcal{F}}$ with the single-step revealing parameter $C_{\mathcal{O}}$.*

*Proof.* Consider the following POMDP: for $1 \leq h \leq H - 1$, there is only one state $s_{h,1}$ (thus transition before $h = H - 1$ is a trivial chain), and $C_{\mathcal{H}} = 1$. At the last step $H$, there are two states

---

[6]Here we assume that the algorithm has knowledge of the value of $C_{\mathcal{F}}$ and $C_{\mathcal{H}}$. The analyses can be easily adapted to the case where the precise value of $C_{\mathcal{F}}$ (or $C_{\mathcal{H}}$) is unknown but a nontrivial upper bound (e.g., a constant multiple of the value) is known.

$s_{H,L}$ and $s_{H,R}$. The emission is identity, i.e., $o_h = s_h$, meaning that the POMDP is also a MDP and $C_{\mathcal{O}} = C_{\mathcal{F}} = 1$. There are two actions, $L$ and $R$. At step $H-1$, taking $L$ and $R$ transitions to $s_{H,L}$ and $s_{H,R}$, respectively. The agent only receives a reward of 1 upon reaching state $s_{H,L}$. The model is known so $|\mathcal{M}| = 1$. The behavior policy $\pi_b$ is uniformly random, implying $C_{\mathcal{A}} = 2$. Consider two target policies, $\pi_1$ and $\pi_2$. Both policies take action $L$ if historical actions do not include $R$ for $1 \le h \le H-2$. At step $H-1$, $\pi_1$ takes action $L$ if all previous actions are $L$, while $\pi_2$ takes action $R$ if all previous actions are $L$. Under any history that includes at least one $R$ action, both policies yield the same action, and the concrete choice does not matter.[7]

$J(\pi_1) = 1$ and $J(\pi_2) = 0$, as running $\pi_1$ produces an all-$L$ sequence, and running $\pi_2$ produces $L, \ldots, L, R$. However, since the two policies have identical action choices under all other action sequences, a model-free algorithm can only tell them apart if the sequence $L, \ldots, L$ of length $H-1$ is observed in the offline data, which only happens with a negligible $O(1/2^H)$ probability. With overwhelming probability, $\pi_1$ and $\pi_2$ will look identical to the algorithm but their $J(\cdot)$ differs by a constant of 1, so no algorithm can predict them well simultaneously up to $\epsilon = 1/2$ accuracy, unless it is given $\Omega(2^H)$ samples to observe the $L, \ldots, L$ sequence with nontrivial probability. However, in this problem, $C_{\mathcal{O}}, C_{\mathcal{F}}, C_{\mathcal{H}}, C_{\mathcal{A}}, \epsilon, |\mathcal{M}|$ are all constants, so the sample complexity in Theorem 1 is $\text{poly}(H)$, which cannot explain away the exponential in $\Omega(2^H)$. Thus we conclude that no model-free algorithm can achieve Theorem 1's guarantee when $\pi_e$ is allowed to be history dependent. □

The counter-intuitive part of the construction is that the model is fully known ($\mathcal{M} = \{M^\star\}$), and the hardness comes merely from the fact that model-free algorithms have limited access to $\pi_e$ and cannot distinguish between two history-dependent policies with very different returns from data. In fact, if we remove the model-free restriction, the problem instance is trivial to solve as we can calculate $J_{M^\star}(\pi_e)$ by rolling out $\pi_e$ in $M^\star$.

## 4 POSITIVE RESULTS FOR MODEL-BASED ALGORITHMS

The negative result from the previous section naturally motivates the investigation of model-based algorithms, i.e., those that do not respect the restriction of Definition 2. Model-based algorithms fit the POMDP dynamics from data and use the learned model to evaluate $\pi_e$; the latter step can often be achieved by rolling out trajectories in the learned model, where we need to query $\pi_e$ on *synthetic* trajectories generated by the learned model, thus violating Definition 2 and potentially circumventing the hardness in Theorem 3.

Concretely, we consider a very natural algorithm based on Maximum Likelihood Estimation (MLE). It is, in fact, quite surprising that such a simple algorithm has not been analyzed under relatively general assumptions for OPE in POMDPs. The algorithm is as follows:

- **Pre-filtering:** We construct $\mathcal{M}' \subset \mathcal{M}$ where all models that violate Assumption E (for Section 4.1 where we assume single-step revealing in the guarantee) or D (for Section 4.2 where we assume multi-step revealing) are excluded from $\mathcal{M}'$.[8] Such a filtering step is common in the POMDP learning literature (Liu et al., 2022a), and in Section 5 we show why MLE estimation requires this as a pre-processing step and will fail otherwise.

- **MLE:** Let $\mathbb{P}_M^\pi$ stands for the probabilities under policy $\pi$ and model $M$. The model is learned as

$$\widehat{M} = \max_{M \in \mathcal{M}'} \sum_{i=1}^n \log \mathbb{P}_M^{\pi_b}(\tau_H^{(i)}). \tag{1}$$

- **Prediction:** We use the expected return of $\pi_e$ in $\widehat{M}$, $J_{\widehat{M}}(\pi_e)$, as our estimation for $J(\pi_e)$.

### 4.1 GUARANTEE UNDER SINGLE-STEP OUTCOME REVEALING

We first show that the model-based algorithm enjoys a polynomial guarantee when we assume the stronger single-step outcome revealing (Assumption E) instead of its multi-step version (Assump-

---

[7]Rolling out $\pi_1$ and $\pi_2$ leads to deterministic outcomes, which are characteristics of open-loop policies (i.e., action only depends on $h$ and is independent of $(\tau_{h-1}, o_h)$). However, $\pi_1$ and $\pi_2$ cannot be open-loop simultaneously, otherwise they cannot have identical action choices on histories that include $R$.

[8]This requires the algorithm to have knowledge of the value or some nontrivial upper bound of $C_{\mathcal{O}}$ (or $C_{\mathcal{F}}$). Such knowledge is also required for Theorem 1.

tion D). Recall that even with this stronger revealing assumption the model-free algorithms still cannot provide polynomial guarantees.

**Theorem 4.** *Given a realizable model class $\mathcal{M}$, let model $\widehat{M}$ be the MLE within $\mathcal{M}'$ using the dataset $\mathcal{D}$. Under Assumption E, with probability at least $1 - \delta$, we have*

$$\left| J(\pi_e) - J_{\widehat{M}}(\pi_e) \right| \leq \mathcal{O}\left( H^2 C_{\mathbb{O}}^2 C_{\text{eff},1} \sqrt{\frac{\log \frac{|\mathcal{M}|}{\delta}}{n}} \right).$$

*where*

$$C_{\text{eff},1} := \max_{h \in [H-1]} \frac{\sum_{o_h, a_h} \mathbb{E}_{\pi_e}\left[ \pi_e(a_h \mid \tau_{h-1}, o_h) \|(\widehat{\mathbf{B}}_h - \mathbf{B}_h)\mathbb{O}_h \mathbf{b}_{\mathcal{S}}(\tau_{h-1})\|_1 \right]}{\sum_{o_h, a_h} \mathbb{E}_{\pi_b}\left[ \pi_b(a_h \mid \tau_{h-1}, o_h) \|(\widehat{\mathbf{B}}_h - \mathbf{B}_h)\mathbb{O}_h \mathbf{b}_{\mathcal{S}}(\tau_{h-1})\|_1 \right]},$$

*Here $\mathbf{B}_h = \mathbf{B}_h(o_h, a_h)$ is the observable-operator parameterization of $M^\star$ under single-step revealing (see Appendix C), and $\widehat{\mathbf{B}}_h$ is defined similarly for $\widehat{M}$. Furthermore, when Assumptions A and C hold, we have*

$$C_{\text{eff},1} \leq C_{\mathcal{A}} C_{\mathcal{H}}.$$

The proof is deferred to Appendix D.1. To the best of our knowledge, this is the first polynomial sample-complexity bound for model-based OPE of history-dependent target policies in POMDPs with large observation spaces. In addition to the quantities that have already been introduced, the bound depends on a new coverage parameter, $C_{\text{eff},1}$, which we show can be upper bounded by $C_{\mathcal{A}} C_{\mathcal{H}}$. With this relaxation, Theorem 4 implies a formal $\text{poly}(H, \log(|\mathcal{M}|/\delta), \epsilon, C_{\mathcal{A}}, C_{\mathcal{F}}, C_{\mathcal{H}})$ sample complexity, in a setting where model-free OPE provably cannot handle (Theorem 3).

However, as mentioned in Section 3.2, the $C_{\mathcal{A}} C_{\mathcal{H}}$ bound is relaxing $C_{\text{eff},1}$ to a "uniform coverage" parameter, whereas $C_{\text{eff},1}$ itself is policy-sepcific (note the dependence on $\pi_e$ on the numerator) and can be much tighter. For starters, when $\pi_e = \pi_b$, it is obvious that $C_{\text{eff},1} = 1$. More generally, $C_{\text{eff},1}$ captures the discrepancy between the model estimation error under the distributions induced by $\pi_e$ and $\pi_b$. Unlike the commonly used $L_2$ norm of Bellman error in offline MDPs (Xie et al., 2021), we use the $L_1$ norm of the estimation error, which better leverages the property of the belief state vector ($\|\mathbf{b}_{\mathcal{S}}(\tau_h)\|_1 = 1$). The advantage of $L_1$ over $L_2$ is also discussed in Zhang & Jiang (2024), where they note that $L_1/L_\infty$ Hölder's inequality can help reduce the dependence on $S$ in the analysis. In the special case of the MDP setting (i.e., $s_h = o_h$ and $\mathbb{O}_h \equiv \mathbf{I}$), we have (see Appendix B)

$$C_{\text{eff},1} \leq \max_h \max_{s_h, a_h} \frac{d_h^{\pi_e}(s_h, a_h)}{d_h^{\pi_b}(s_h, a_h)}.$$

This recovers a coverage coefficient in the form of state-action density ratio, which is commonly used in the offline MDP literature (Munos, 2007; Antos et al., 2008; Chen & Jiang, 2019; Jiang & Xie, 2024).

## 4.2 Guarantee under Multi-step Outcome Revealing

Theorem 4 relies on Assumption E, which requires that the observation reveals sufficient information about the latent state. However, this assumption may not hold in more complex partially observable environments. We now show that the model-based algorithm can also enjoy polynomial guarantees under the weaker multi-step revealing condition (Assumption D), if we impose that the behavior policy $\pi_b$ is memoryless (the target policy $\pi_e$ is still history dependent).

**Theorem 5.** *Given a realizable model class $\mathcal{M}$, let model $\widehat{M}$ be the MLE within $\mathcal{M}'$ using dataset $\mathcal{D}$. Suppose $\pi_b$ is memoryless and Assumption D hold, with probability at least $1 - \delta$, we have*

$$\left| J(\pi_e) - J_{\widehat{M}}(\pi_e) \right| \leq \mathcal{O}\left( H^2 C_{\mathcal{F}}^2 C_{\text{eff},m} \sqrt{\frac{\log \frac{|\mathcal{M}|}{\delta}}{n}} \right).$$

*where*

$$C_{eff,m} := \max_{h \in [H-1]} \frac{\sum_{o_h, a_h} \mathbb{E}_{\pi_e} \left[ \pi_e(a_h \mid \tau_{h-1}, o_h) \| (\widehat{\mathbf{B}}_h - \mathbf{B}_h) U_{\mathcal{F}, h} \mathbf{b}_{\mathcal{S}}(\tau_{h-1}) \|_1 \right]}{\sum_{o_h, a_h} \mathbb{E}_{\pi_b} \left[ \pi_b(a_h \mid \tau_{h-1}, o_h) \| (\widehat{\mathbf{B}}_h - \mathbf{B}_h) U_{\mathcal{F}, h} \mathbf{b}_{\mathcal{S}}(\tau_{h-1}) \|_1 \right]}, \tag{2}$$

*where* $\mathbf{B}_h = \mathbf{B}_h(o_h, a_h)$ *is the observable-operator parameterization of* $M^\star$ *under multi-step revealing (Appendix C). Furthermore, similar to the single-step case, when Assumption C holds we can upper bound* $C_{eff,m}$ *as*

$$C_{eff,m} \leq C_{\mathcal{A}} C_{\mathcal{H}}.$$

The proof is deferred to Appendix D.2. Our theorem shows that when the behavior policy $\pi_b$ is memoryless, only polynomial samples are required to evaluate the target policy $\pi_e$ under Assumption D. The structure of the bound is similar to that of Theorem 4, except that it replaces $C_{\mathcal{O}}$ with its multi-step counterpart $C_{\mathcal{F}}$.

Interestingly, our results reveal that a memoryless behavior policy can accurately evaluate history-dependent target policies when certain coverage conditions are satisfied. This is somewhat counter-intuitive, as one might expect that only history-dependent behavior policies could evaluate history-dependent target policies. However, this conclusion aligns with findings in latent MDPs (Kwon et al., 2024), which are a special case of POMDPs. This insight encourages the use of memoryless policies with good coverage for online exploration, consistent with many online algorithms that employ uniform exploration policies (Efroni et al., 2022; Liu et al., 2022a; Uehara et al., 2022a; Guo et al., 2023b).

## 5 STATE-SPACE MISMATCH AND MODEL MISSPECIFICATION

So far we assume the agent is given a realizable model class with the ground-truth latent-state space size $|\mathcal{S}|$. Although the knowledge of $|\mathcal{S}|$ is commonly assumed in both OPE in POMDPs (Uehara et al., 2023a; Zhang & Jiang, 2024) and online learning for POMDPs (Jin et al., 2020; Liu et al., 2022a), obtaining this information in practice can be challenging, as the latent state space is an ungrounded object. A natural concern is that of state-space misspecification (Kulesza et al., 2014; 2015): what if the state space of models in $\mathcal{M}$ (denoted as $\widehat{\mathcal{S}}$ in this section) is different from that of $M^\star$? This question has largely eluded the recent literature on RL in POMDPs, and below we show that taking it seriously reveals interesting insights.

One immediate consequence of $|\widehat{\mathcal{S}}| \neq |\mathcal{S}|$ is that the model realizability assumption $M^\star \in \mathcal{M}$ no longer makes any sense. A natural rescue is to note that it is still possible to have models in $\mathcal{M}$ that induce the same *observable process* as $M^\star$, that is, they always generate the same distribution of observation-action sequences given any policy. For example, if $|\widehat{\mathcal{S}}| > |\mathcal{S}|$, we can still hope this weaker notion of realizability (based on observable equivalence) holds, since one can simply add dummy latent states to $M^\star$ to create an equivalent model with larger latent-state spaces. More concretely:

**Definition 6.** We say that $\mathcal{M}$ satisfies *observable-equivalent* realizability, if there exists $M \in \mathcal{M}$, such that $P_M(o_{h+1}|o_1, a_1, \ldots, o_h, a_h)$ is the same as $P_{M^\star}(o_{h+1}|o_1, a_1, \ldots, o_h, a_h)$ for all $h$ and $o_1, a_1, \ldots, o_{h+1}$.

We now show a somewhat counter-intuitive result: replacing $M^\star \in \mathcal{M}$ with observable-equivalent realizability will break the previous guarantee (e.g., Theorem 4).

**Theorem 7.** *Take the same setting as Theorem 4, with model realizability (Assumption B) replaced by the weaker observable-equivalent realizability (Definition 6). The model-based algorithm, either with or without the pre-filtering step (Section 4), cannot achieve the polynomial guarantee in Theorem 4.*

*Proof.* The action space is $\{L, R\}$ and rewards are only received at final step $H$. $\mathcal{M} = \{M_1, M_2\}$, i.e., there are only two candidate models in $\mathcal{M}$. All the models including $M^\star$ share the same observation space, which contains only one observation for $h \in [H-1]$ and two observations at step $H$. $M_1$ and $M_2$ have the same state space as $M^\star$ at the final step $H$: $s_{\text{good}}$ and $s_{\text{bad}}$, state $s_{\text{good}}$ always generates $o_{\text{good}}$ with reward 1 and state $s_{\text{bad}}$ always generates $o_{\text{bad}}$ with reward 0. In $M_1$,

the state space at step $h \in [H-1]$ is $\{L, R\}^{h-1}$. For $h \in [H-2]$, any state-action pair $(s_h, a_h)$ deterministically transitions to the next state $s_{h+1} = (s_h, a_h)$, where $a_h$ is appended to $s_h$. At step $H-1$, all states transition to $s_{\text{good}}$ when action $L$ is taken; otherwise, they transition to $s_{\text{bad}}$. $M_2$ shares the same state space and dynamics as $M_1$, except that at step $H-1$, the all-$R$ state will transition to $s_{\text{good}}$ if action $R$ is taken. The ground-truth model $M^\star$ has only one state at each step $h \in [H-1]$. At step $H-1$, it transitions to $s_{\text{good}}$ if action $L$ is taken; otherwise, it transitions to $s_{\text{bad}}$. We can verify that $M_1$ produces the same observable process as $M^\star$. For $M^\star$, since there is only one state and one observation for $h \in [H-1]$, we have $C_{\mathcal{O}} = C_{\mathcal{H}} = 1$. The behavior policy $\pi_b$ is uniformly random with $C_{\mathcal{A}} = 2$, while the target policy $\pi_e$ always takes action $R$.

We first consider the case where the pre-filtering step is skipped, so both $M_1$ and $M_2$ are considered for MLE. To estimate $\pi_e$ accurately, the algorithm needs to output $\widehat{M} = M_1$ ($M_2$'s prediction is wrong by a constant), but the empirical MLE losses (Eq.(1)) of $M_1$ and $M_2$ are always identical unless the all-$R$ action sequence is contained in the dataset . Since $\pi_b$ is uniformly random, $\Omega(2^H)$ samples are needed to include this sequence with nontrivial probability. Therefore, with only $\text{poly}(H)$ samples, MLE cannot distinguish between $M_1$ and $M_2$, and hence cannot accurately estimate $J(\pi_e)$.

If the algorithm does perform the pre-filtering step, note that $C_{\mathcal{O}} = \infty$ for both models as $|\mathcal{S}_h| > |\mathcal{O}_h|$ for $h > 1$ and $\Sigma_{\mathcal{O}, h}$ is non-invertible, so both models will be eliminated and the algorithm cannot be executed. $\qquad\square$

In the model-based algorithm in Section 4, we assume revealing assumptions (Assumptions E or D) on $M^\star$ and perform pre-filtering. The negative result here sheds light on this and reveals the underlying reason: what we really need is not $M^\star$, but the learned model $\widehat{M}$ to satisfy revealing properties. In the proof of Theorem 7, no models in $\mathcal{M}$ satisfy single-step revealing (Assumption E) which leads to poor sample efficiency. Inspired by this observation, we present the following result that handles state-space mismatch properly. We assume $|\widehat{\mathcal{S}}|$ can be either smaller or greater than $|\mathcal{S}|$, and an approximate version of observable-equivalent realizability.

**Theorem 8.** *Given a model class $\mathcal{M}$ such that each $M \in \mathcal{M}$ satisfies Assumption D. Let model $\widehat{M}$ be the MLE within $\mathcal{M}$ using dataset $\mathcal{D}$. Suppose $\pi_b$ is memoryless, with probability at least $1 - \delta$,*

$$\left| J(\pi_e) - J_{\widehat{M}}(\pi_e) \right| \leq \mathcal{O}\left( H^2 C_{\mathcal{F}}^2 C_{eff, m} \sqrt{\frac{\log \frac{|\mathcal{M}|}{\delta}}{n}} + \varepsilon_{approx} \right),$$

*where $C_{eff, m}$ is defined in Eq. (2) and $\varepsilon_{approx}$ is the approximation error of $\mathcal{M}$:*

$$\varepsilon_{approx} := \min_{M \in \mathcal{M}} \frac{1}{n} \sum_{i=1}^{n} \left( \log \mathbb{P}_{M^\star}^\pi (\tau_H^{(i)}) - \log \mathbb{P}_M^\pi (\tau_H^{(i)}) \right).$$

The proof is deferred to Appendix E. Here we assume that all $M \in \mathcal{M}$ satisfies Assumption D. It is actually fine if $M^\star$ itself does not satisfy the assumption, as long as one of models $M \in \mathcal{M}$ is observable-equivalent to $M^\star$ up to approximation error $\varepsilon_{\text{approx}}$.

## 6 RELATED WORKS

**OPE in POMDPs.** There are two main directions in the study of OPE in POMDPs: OPE in confounded POMDPs and OPE in unconfounded POMDPs. OPE in confounded POMDPs (Zhang & Bareinboim, 2016; Namkoong et al., 2020; Tennenholtz et al., 2020; Nair & Jiang, 2021; Shi et al., 2022; Xu et al., 2023; Bennett & Kallus, 2024) assume that the actions from the behavior policy depend only on the latent state. As a result, these methods are inapplicable to our unconfounded setting, where the latent state is unobservable, and the behavior policy depends solely on the observations and actions. A line of research (Hu & Wager, 2023; Uehara et al., 2023a; Zhang & Jiang, 2024) investigates the same unconfounded setting as ours. Hu & Wager (2023) employ multi-step importance sampling, which leads to an exponential dependence on the horizon length. Uehara et al. (2023a); Zhang & Jiang (2024) use model-free methods that estimate the future-dependent value functions of the target policy, but their approaches focus on memoryless target policies or policies with limited memory. In this paper, we leverage model-based methods and provide polynomial sample complexity bounds for history-dependent target policies.

**OPE in MDPs.** OPE is an important problem in MDPs and many works have studied it, including importance sampling methods (Precup, 2000; Li et al., 2011), marginalized importance sampling approaches (Liu et al., 2018; Xie et al., 2019; Kallus & Uehara, 2020; Katdare et al., 2023), doubly robust estimators (Dudík et al., 2011; Thomas & Brunskill, 2016; Jiang & Li, 2016; Farajtabar et al., 2018; Xu et al., 2021) and model-based methods (Eysenbach et al., 2020; Voloshin et al., 2021; Yin et al., 2021). However, most of these methods (except for importance sampling) require the Markovian property of the environment. Applying these techniques to POMDPs either does not work or results in an exponential dependence on the horizon length. Xie et al. (2019) demonstrated that state-action coverage is sufficient to achieve polynomial sample complexity in MDPs. However, as shown by Kwon et al. (2024), latent state coverage is insufficient in POMDPs due to their partial observability. In this paper, we consider coverage conditions for both histories and outcomes, a direction that is also explored in Zhang & Jiang (2024) for model-free methods.

**Online Learning in POMDPs.** Online learning algorithms in POMDPs have been extensively studied. Krishnamurthy et al. (2016) show that the lower bound on the sample complexity of learning general POMDPs is exponential in the horizon, and later works have focused on circumventing the hardness with additional assumptions. Uehara et al. (2023b) propose provably efficient algorithms for POMDPs with deterministic transition dynamics. Guo et al. (2016); Azizzadenesheli et al. (2016); Xiong et al. (2022b) assume the environment satisfies the reachability assumption or that exploratory data is available. There are also many works studying sub-classes of POMDPs, including latent MDPs (Kwon et al., 2021; 2024), decodable POMDPs (Krishnamurthy et al., 2016; Jiang et al., 2017; Du et al., 2019; Efroni et al., 2022), weakly revealing POMDPs (Jin et al., 2020; Liu et al., 2022a;b), low-rank POMDPs (Wang et al., 2022; Guo et al., 2023b), POMDPs with hindsight observability (Lee et al., 2023; Guo et al., 2023a) and observable POMDPs (Golowich et al., 2022a;b), except for the model-free algorithm of Uehara et al. (2022a) based on the future-dependent value function framework (Uehara et al., 2023a).

## 7 CONCLUSION

We studied OPE of history-dependent target policies in POMDPs with large observation spaces, and showed provable separation between model-free and model-based methods in several settings. A major open problem is whether model-based algorithm can handle history-dependent $\pi_b$ and multi-step revealing (the "?" mark in Table 1), and resolving this question will provide a more comprehensive picture of the landscape of OPE in POMDPs.

## ACKNOWLEDGMENTS

Nan Jiang acknowledges funding support from NSF CNS-2112471, NSF CAREER IIS-2141781, Google Scholar Award, and Sloan Fellowship.

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

## A    PROOF OF THEOREM 1

*Proof.* We use the algorithm described in Section 3 of Zhang & Jiang (2024). Given the model class $\mathcal{M}$, we construct the function classes $\mathcal{V}$ and $\Xi$ needed by their algorithm. First, we apply the same pre-filtering as in the model-based algorithm to obtain $\mathcal{M}'$. For each $M \in \mathcal{M}'$, we include the corresponding future-dependent value function (FDVF) $V_{\mathcal{F}}$ (their Definition 3) in $\mathcal{V}$. For each model pair $M_1, M_2 \in \mathcal{M}'$, we include $\mathcal{B}_{M_1}^{\mathcal{H}} V_{\mathcal{F},M_2}$ (their Definition 4) in $\Xi$, where $V_{\mathcal{F},M_2}$ is the FDVF in model $M_2$ and $\mathcal{B}_{M_1}^{\mathcal{H}}$ is the Bellman residual operator under model $M_1$. Hence, we have $|\mathcal{V}| \leq |\mathcal{M}|$ and $|\Xi| \leq |\mathcal{M}|^2$. Meanwhile, their Assumptions 5 and 6 are naturally satisfied. From our Assumption C, their Assumption 11 is also satisfied with $C_{\mathcal{H},2} \leq C_{\mathcal{H}}^2$. According to their Lemma 5, we have $\|V_{\mathcal{F}}\|_\infty \leq HC_{\mathcal{F}}$. Then, we invoke their Theorem 7 and obtain that w.p. $1 - \delta$,

$$|J(\pi_e) - \widehat{J}(\pi_e)| \leq \mathcal{O}\left( H^2 C_{\mathcal{F}} C_{\mathcal{H}} \sqrt{\frac{C_{\mathcal{A}} \log(|\mathcal{M}|/\delta)}{n}} \right). \qquad \square$$

## B    CALCULATION OF THE MDP CASE

For MDP, we have $o_h = s_h$ and $\mathbb{O}_h \equiv I$. Hence, given $\tau_{h-1}, s_h, a_h$,

$$\pi_e(a_h \mid \tau_{h-1}, s_h) \left\| \left( \mathbf{B}_h(s_h, a_h) - \widehat{\mathbf{B}}_h(s_h, a_h) \right) \mathbb{O}_h \mathbf{b}_{\mathcal{S}}(\tau_{h-1}) \right\|_1$$

$$= \pi_e(a_h \mid \tau_{h-1}, s_h) \left\| (\mathbb{T}_{h,a_h} - \widehat{\mathbb{T}}_{h,a_h}) \operatorname{diag}(\mathbb{O}_h(s_h \mid \cdot)) \mathbf{b}_{\mathcal{S}}(\tau_{h-1}) \right\|_1$$

$$= \pi_e(a_h \mid \tau_{h-1}, s_h) \sum_{s'} \mathbb{P}_{M^\star}(s_h \mid \tau_{h-1}) \left| \mathbb{P}_{\widehat{M}}(s' \mid s_h, a_h) - \mathbb{P}_{M^\star}(s' \mid s_h, a_h) \right|$$

$$= \mathbb{P}_{M^\star}^{\pi_e}(s_h, a_h \mid \tau_{h-1}) \left\| \mathbb{P}_{\widehat{M}}(\cdot \mid s_h, a_h) - \mathbb{P}_{M^\star}(\cdot \mid s_h, a_h) \right\|_1 .$$

Fix step $h$, the numerator in $C_{\mathrm{eff},1}$ becomes

$$\sum_{s_h, a_h} d_h^{\pi_e}(s_h, a_h) \left\| \mathbb{P}_{\widehat{M}}(\cdot \mid s_h, a_h) - \mathbb{P}_{M^\star}(\cdot \mid s_h, a_h) \right\|_1 .$$

Similarly, the denominator is

$$\sum_{s_h, a_h} d_h^{\pi_b}(s_h, a_h) \left\| \mathbb{P}_{\widehat{M}}(\cdot \mid s_h, a_h) - \mathbb{P}_{M^\star}(\cdot \mid s_h, a_h) \right\|_1 .$$

Then we have

$$C_{\mathrm{eff},1} \leq \max_h \max_{s_h, a_h} \frac{d_h^{\pi_e}(s_h, a_h)}{d_h^{\pi_b}(s_h, a_h)}.$$

## C    OBSERVABLE OPERATOR MODELS

We introduce the observable operator models (OOMs) (Jaeger, 2000), which (for our purposes) can be viewed as a reparameterization of POMDPs. OOMs have seen wide use in recent POMDP literature (especially in the online setting (Liu et al., 2022a)), and play a crucial role in our model-based analyses.

**Single-step Revealing.**    We first introduce the OOM used in environments satisfying single-step revealing condition. Given model parameters $M^\star = (\mathbb{T}, \mathbb{O}, d_1)$, we write the initial observation distribution $\mathbf{b}_0$ and observable operators $\mathbf{B}$ as

$$\mathbf{b}_0 = \mathbb{O}_1 d_1 \in \mathbb{R}^O,$$

$$\mathbf{B}_h(o, a) = \mathbb{O}_{h+1} \mathbb{T}_{h,a} \operatorname{diag}(\mathbb{O}_h(o \mid \cdot)) \mathbb{O}_h^{\dagger,w} \in \mathbb{R}^{O \times O},$$

where $\mathbb{T}_{h,a} \in \mathbb{R}^{|\mathcal{S}_{h+1}| \times |\mathcal{S}_h|}$ with $[\mathbb{T}_{h,a}]_{ij} = \mathbb{T}_h(s_{h+1} = i \mid s_h = j, a_h = a)$. Different from the pseudo-inverse in Liu et al. (2022a), we use the weighted version (Zhang & Jiang, 2024) defined as follows:

$$\mathbb{O}_h^{\dagger,w} = \Sigma_{\mathcal{O},h}^{-1} \mathbb{O}_h^\top W_h^{-1},$$

where $\Sigma_{\mathcal{O},h}$ and $W_h$ are defined in Assumption E. With the OOM representaion, for any trajectory $\tau_h = (o_1, a_1, \cdots, o_h, a_h)$, the probability of generating $\tau_h$ with policy $\pi$ can be written as

$$\mathbb{P}_{M^\star}^\pi(\tau_h) = \pi(\tau_h) \cdot \left(\mathbf{e}_{o_h}^\top \mathbf{B}_{h-1}(o_{h-1}, a_{h-1}) \cdots \mathbf{B}_1(o_1, a_1)\mathbf{b}_0\right),$$

where $\pi(\tau_h) = \prod_{h'=1}^h \pi(a_h \mid o_h, \tau_{h-1})$ is the action probability of generating $\tau_h$ given policy $\pi$. Moreover, let $\mathbf{b}(\tau_h) = \left(\prod_{h'=1}^h \mathbf{B}_{h'}(o_{h'}, a_{h'})\right)\mathbf{b}_0$, we have the following observation:

$$\mathbf{b}(\tau_h) = \mathbb{P}_{M^\star}(\tau_h)\mathbb{O}_{h+1}\mathbf{b}_{\mathcal{S}}(\tau_h),$$

where $\mathbb{P}_{M^\star}(\tau_h) = \mathbf{e}_{o_h}^\top \mathbf{B}_{h-1}(o_{h-1}, a_{h-1}) \cdots \mathbf{B}_1(o_1, a_1)\mathbf{b}_0$ is the environment probability of generating $\tau_h$. The relationship between $\mathbf{b}(\tau_h)$ and belief state vector $\mathbf{b}_{\mathcal{S}}(\tau_h)$ is crucial for developing coverage conditions on the belief state.

**Multi-step Revealing.**   Next, we introduce the OOM in the multi-step revealing setting. Recall that $f_h$ denotes the future after step $h$, and $U_{\mathcal{F},h}$ is the outcome matrix, where the row indexed by $f_h$ represents its corresponding outcome vector. The observable operators are defined as

$$\mathbf{b}_0 = U_{\mathcal{F},1}d_1 \in \mathbb{R}^{|\mathcal{F}_1|},$$
$$\mathbf{B}_h(o, a) = U_{\mathcal{F},h+1}\mathbb{T}_{h,a}\mathrm{diag}(\mathbb{O}_h(o \mid \cdot))U_{\mathcal{F},h}^{\dagger,w} \in \mathbb{R}^{|\mathcal{F}_{h+1}| \times |\mathcal{F}_h|}.$$

Here $U_{\mathcal{F},h}^{\dagger,w} = \Sigma_{\mathcal{F},h}^{-1} U_{\mathcal{F},h}^\top Z_h^{-1}$ is the weighted pseudo-inverse and $\Sigma_{\mathcal{F},h}$, $Z_h$ are defined in Assumption D. For any history $\tau_{h-1}$ and future $f_h$, the probability of generating $(\tau_{h-1}, f_h)$ with policy $\pi$ is written as

$$\mathbb{P}_{M^\star}^\pi(\tau_{h-1}, f_h) = \pi(\tau_{h-1}) \cdot \left(\mathbf{e}_{f_h}^\top \mathbf{B}_{h-1}(o_{h-1}, a_{h-1}) \cdots \mathbf{B}_1(o_1, a_1)\mathbf{b}_0\right),$$

Similar to the single-step case, we also have the following relation between $\mathbf{b}(\tau_h)$ and $\mathbf{b}_{\mathcal{S}}(\tau_h)$:

$$\mathbf{b}(\tau_h) = \mathbb{P}_{M^\star}(\tau_h)U_{\mathcal{F},h+1}\mathbf{b}_{\mathcal{S}}(\tau_h),$$

# D  PROOFS FOR SECTION 4

We first provide a lemma showing that a model can have accurate probability estimation of trajectories when it estimates the historical data well. The proof can be found in Appendix B of Liu et al. (2022a).

**Lemma 9** (Restatement of Proposition 14 of Liu et al. (2022a)). *There exists a universal constant $c$ such that for any $\delta \in (0, 1]$, with probability at least $1 - \delta$, for all $M \in \mathcal{M}$, it holds that*

$$\left(\sum_{\tau \in \prod_{h=1}^H (\mathcal{O}_h \times \mathcal{A})} |\mathbb{P}_M^{\pi_b}(\tau) - \mathbb{P}_{M^\star}^{\pi_b}(\tau)|\right)^2 \leq \frac{c\left(\sum_{i=1}^n \log \frac{\mathbb{P}_{M^\star}^{\pi_b}(\tau^{(i)})}{\mathbb{P}_M^{\pi_b}(\tau^{(i)})} + \log \frac{|\mathcal{M}|}{\delta}\right)}{n}$$

Since the ground-truth $M^\star \in \mathcal{M}$, for the MLE $\widehat{M}$, we further have

$$\sum_{\tau \in \prod_{h=1}^H (\mathcal{O}_h \times \mathcal{A})} \left|\mathbb{P}_{\widehat{M}}^{\pi_b}(\tau) - \mathbb{P}_{M^\star}^{\pi_b}(\tau)\right| \leq \sqrt{\frac{c\log \frac{|\mathcal{M}|}{\delta}}{n}} := \varepsilon_{\mathrm{MLE}}.$$

In addition, marginalizing two distributions will not increase the TV distance, for any $h \in [H-1]$, the following inequalities hold,

$$\sum_{\tau_h, o_{h+1}} \left|\mathbb{P}_{\widehat{M}}^{\pi_b}(\tau_h, o_{h+1}) - \mathbb{P}_{M^*}^{\pi_b}(\tau_h, o_{h+1})\right| \leq \varepsilon_{\mathrm{MLE}}, \tag{3}$$

$$\sum_{\tau_h, f_{h+1}} \left|\mathbb{P}_{\widehat{M}}^{\pi_b}(\tau_h, f_{h+1}) - \mathbb{P}_{M^*}^{\pi_b}(\tau_h, f_{h+1})\right| \leq \varepsilon_{\mathrm{MLE}}. \tag{4}$$

### D.1 PROOF OF THEOREM 4

To begin with, we first state the following lemma for our operators $\mathbf{B}$, which is adapted from Lemma 32 of Liu et al. (2022a). We use $\mathbf{B}_{h:j+1}$ to denote $\prod_{h'=j+1}^{h} \mathbf{B}_{h'}(o_{h'}, a_{h'})$.

**Lemma 10.** *For any index $0 \leq j < h \leq H - 1$, trajectory $\tau_j \in \prod_{h'=1}^{j}(\mathcal{O}_{h'} \times \mathcal{A})$, vector $x \in \mathbb{R}^O$, policy $\pi$, and operator $\mathbf{B}$ satisfying Assumption E, we have*

$$\sum_{\tau_{h:j+1}} \|\mathbf{B}_{h:j+1} x\|_1 \times \pi(\tau_{h:j+1} \mid \tau_j) \leq C_{\mathcal{O}} \|x\|_1.$$

*Proof.* From the definition of $\mathbf{B}_{h:j+1}$, we have

$$\mathbf{B}_{h:j+1} x = \mathbf{B}_{h:j+1} \mathbb{O}_{j+1} \mathbb{O}_{j+1}^{\dagger,w} x.$$

Then, for any standard basis $\mathbf{e}_i \in \mathbb{R}^S$, we obtain

$$\sum_{\tau_{h:j+1}} \|\mathbf{B}_{h:j+1} \mathbb{O}_{j+1} \mathbf{e}_i\|_1 \times \pi(\tau_{h:j+1} \mid \tau_j)$$

$$= \sum_{o} \sum_{\tau_{h:j+1}} \mathbb{P}_M(o_{h+1} = o, \tau_{h:j+1} \mid s_{j+1} = i) \cdot \pi(\tau_{h:j+1} \mid \tau_j)$$

$$= \sum_{o} \sum_{\tau_{h:j+1}} \mathbb{P}_M^{\pi}(o_{h+1} = o, \tau_{h:j+1} \mid \tau_j, s_{j+1} = i) = 1.$$

Therefore,

$$\sum_{\tau_{h:j+1}} \|\mathbf{B}_{h:j+1} x\|_1 \times \pi(\tau_{h:j+1} \mid \tau_j)$$

$$\leq \sum_{\tau_{h:j+1}} \|\mathbf{B}_{h:j+1} \mathbb{O}_{j+1}\|_1 \|\mathbb{O}_{j+1}^{\dagger,w} x\|_1 \times \pi(\tau_{h:j+1} \mid \tau_j)$$

$$\leq \|\mathbb{O}_{j+1}^{\dagger,w} x\|_1 \leq C_{\mathcal{O}} \|x\|_1.$$

The last step is from Assumption E and $\|\mathbb{O}_{j+1}^{\dagger,w}\|_1 \leq \|\Sigma_{\mathcal{O},h}^{-1}\|_1 \|\mathbb{O}_h^{\top} W_h^{-1}\|_1 \leq C_{\mathcal{O}}$. $\qquad\square$

Then, we prove Theorem 4. When the context is clear, we omit the $(o_h, a_h)$ in $\mathbf{B}_h(o_h, a_h)$.

*Proof.* We first bound the single-step estimation error of $\pi_b$. For any $h \in [H - 1]$, we have

$$\sum_{\tau_h} \pi_b(\tau_h) \times \left\| \left( \widehat{\mathbf{B}}_h(o_h, a_h) - \mathbf{B}_h(o_h, a_h) \right) \mathbf{b}(\tau_{h-1}) \right\|_1$$

$$\leq \sum_{\tau_h} \pi_b(\tau_h) \times \left\| \widehat{\mathbf{B}}_h(o_h, a_h) \widehat{\mathbf{b}}(\tau_{h-1}) - \mathbf{B}_h(o_h, a_h) \mathbf{b}(\tau_{h-1}) \right\|_1$$

$$+ \sum_{\tau_h} \pi_b(\tau_h) \times \left\| \widehat{\mathbf{B}}_h(o_h, a_h) \left( \widehat{\mathbf{b}}(\tau_{h-1}) - \mathbf{b}(\tau_{h-1}) \right) \right\|_1$$

$$\leq 2 C_{\mathcal{O}} \varepsilon_{\mathrm{MLE}}. \tag{5}$$

For the last step, the first term is bounded by $\varepsilon_{\mathrm{MLE}}$ due to Eq. (3). The second term is from

$$\sum_{\tau_h} \pi_b(\tau_h) \times \left\| \widehat{\mathbf{B}}_h(o_h, a_h) \left( \widehat{\mathbf{b}}(\tau_{h-1}) - \mathbf{b}(\tau_{h-1}) \right) \right\|_1$$

$$= \sum_{\tau_{h-1}} \pi_b(\tau_{h-1}) \sum_{o_h, a_h} \pi_b(a_h \mid \tau_{h-1}, o_h) \left\| \widehat{\mathbf{B}}_h(o_h, a_h) \left( \widehat{\mathbf{b}}(\tau_{h-1}) - \mathbf{b}(\tau_{h-1}) \right) \right\|_1$$

$$\leq C_{\mathcal{O}} \sum_{\tau_{h-1}} \pi_b(\tau_{h-1}) \left\| \widehat{\mathbf{b}}(\tau_{h-1}) - \mathbf{b}(\tau_{h-1}) \right\|_1 \qquad \text{(Lemma 10)}$$

$$\leq C_{\mathcal{O}} \varepsilon_{\mathrm{MLE}}. \qquad \text{(Eq. (3))}$$

Then we consider the evaluation error of $\pi_e$. We decompose the evaluation error into single-step error

$$
\begin{aligned}
\left| J(\pi_e) - J_{\widehat{M}}(\pi_e) \right| &= \sum_{h=1}^{H} \left| \sum_{o_h} (\mathbb{P}_{\widehat{M}}^{\pi_e}(o_h) - \mathbb{P}_{M^*}^{\pi_e}(o_h)) r(o_h) \right| \\
&\leq \sum_{h=1}^{H} \sum_{o_h} \left| \mathbb{P}_{\widehat{M}}^{\pi_e}(o_h) - \mathbb{P}_{M^*}^{\pi_e}(o_h) \right| \\
&= \sum_{h=1}^{H} \sum_{o_h} \left| \sum_{\tau_{h-1}} (\mathbb{P}_{\widehat{M}}^{\pi_e}(o_h, \tau_{h-1}) - \mathbb{P}_{M^*}^{\pi_e}(o_h, \tau_{h-1})) \right| \\
&= \sum_{h=1}^{H} \sum_{o_h} \left| \mathbf{e}_{o_h}^{\top} \left( \sum_{\tau_{h-1}} \left( \widehat{\mathbf{B}}_{h-1} \cdots \widehat{\mathbf{B}}_1 \widehat{\mathbf{b}}_0 - \mathbf{B}_{h-1} \cdots \mathbf{B}_1 \mathbf{b}_0 \right) \times \pi_e(\tau_{h-1}) \right) \right| \\
&= \sum_{h=1}^{H} \left\| \sum_{\tau_{h-1}} \left( \widehat{\mathbf{B}}_{h-1} \cdots \widehat{\mathbf{B}}_1 \widehat{\mathbf{b}}_0 - \mathbf{B}_{h-1} \cdots \mathbf{B}_1 \mathbf{b}_0 \right) \times \pi_e(\tau_{h-1}) \right\|_1 .
\end{aligned}
$$

For the case $h = 1$, according to Eq. (3), we have $\sum_{o_1} \left| \mathbb{P}_{\widehat{M}}^{\pi_e}(o_1) - \mathbb{P}_{M^*}^{\pi_e}(o_1) \right| \leq \varepsilon_{\mathrm{MLE}}$. For $2 \leq h \leq H$, we use the telescoping and obtain the following equality for any $\tau_{h-1}$.

$$
\widehat{\mathbf{B}}_{h-1} \cdots \widehat{\mathbf{B}}_1 \widehat{\mathbf{b}}_0 - \mathbf{B}_{h-1} \cdots \mathbf{B}_1 \mathbf{b}_0 = \sum_{j=1}^{h-1} \widehat{\mathbf{B}}_{h-1:j+1} (\widehat{\mathbf{B}}_j - \mathbf{B}_j) \mathbf{b}(\tau_{j-1}) + \widehat{\mathbf{B}}_{h-1:1} (\widehat{\mathbf{b}}_0 - \mathbf{b}_0). \quad (6)
$$

Therefore, for a fixed step $2 \leq h \leq H$, we have

$$
\begin{aligned}
& \left\| \sum_{\tau_{h-1}} \left( \widehat{\mathbf{B}}_{h-1} \cdots \widehat{\mathbf{B}}_1 \widehat{\mathbf{b}}_0 - \mathbf{B}_{h-1} \cdots \mathbf{B}_1 \mathbf{b}_0 \right) \times \pi_e(\tau_{h-1}) \right\|_1 . \\
\leq & \left\| \sum_{\tau_{h-1}} \pi_e(\tau_{h-1}) \widehat{\mathbf{B}}_{h-1:1} (\widehat{\mathbf{b}}_0 - \mathbf{b}_0) \right\|_1 + \left\| \sum_{\tau_{h-1}} \pi_e(\tau_{h-1}) \sum_{j=1}^{h-1} \widehat{\mathbf{B}}_{h-1:j+1} (\widehat{\mathbf{B}}_j - \mathbf{B}_j) \mathbf{b}(\tau_{j-1}) \right\|_1 .
\end{aligned}
$$

For the first term, we have

$$
\begin{aligned}
\left\| \sum_{\tau_{h-1}} \pi_e(\tau_{h-1}) \widehat{\mathbf{B}}_{h-1:1} (\widehat{\mathbf{b}}_0 - \mathbf{b}_0) \right\|_1 &\leq \sum_{\tau_{h-1}} \left\| \widehat{\mathbf{B}}_{h-1:1} (\widehat{\mathbf{b}}_0 - \mathbf{b}_0) \right\|_1 \times \pi_e(\tau_{h-1}) \\
&\leq C_{\mathcal{O}} \| \widehat{\mathbf{b}}_0 - \mathbf{b}_0 \|_1 && \text{(Lemma 10)} \\
&\leq C_{\mathcal{O}} \varepsilon_{\mathrm{MLE}}. && \text{(Eq. (3))}
\end{aligned}
$$

For the second term, we have

$$
\begin{aligned}
& \left\| \sum_{\tau_{h-1}} \pi_e(\tau_{h-1}) \sum_{j=1}^{h-1} \widehat{\mathbf{B}}_{h-1:j+1} (\widehat{\mathbf{B}}_j - \mathbf{B}_j) \mathbf{b}(\tau_{j-1}) \right\|_1 \\
= & \left\| \sum_{j=1}^{h-1} \sum_{\tau_j} \pi_e(\tau_j) \sum_{\tau_{h-1:j+1}} \pi_e(\tau_{h-1:j+1} \mid \tau_j) \times \widehat{\mathbf{B}}_{h-1:j+1} (\widehat{\mathbf{B}}_j - \mathbf{B}_j) \mathbf{b}(\tau_{j-1}) \right\|_1 \\
= & \left\| \sum_{j=1}^{h-1} \sum_{\tau_j} \pi_e(\tau_j) \mathbf{P}_{\tau_j} \widehat{\mathbb{O}}_{j+1}^{\dagger, w} (\widehat{\mathbf{B}}_j - \mathbf{B}_j) \mathbf{b}(\tau_{j-1}) \right\|_1 , \quad (7)
\end{aligned}
$$

where

$$
\mathbf{P}_{\tau_j} = \sum_{\tau_{h-1:j+1}} \pi_e(\tau_{h-1:j+1} \mid \tau_j) \times \widehat{\mathbf{B}}_{h-1:j+1} \widehat{\mathbb{O}}_{j+1}
$$

and we observe that

$$[\mathbf{P}_{\tau_j}]_{ik} = \sum_{\tau_{h-1:j+1}} \pi_e(\tau_{h-1:j+1} \mid \tau_j) \mathbb{P}_{\widehat{M}}(o_h = i, o_{h-1:j+1} \mid s_{j+1} = k, a_{h:j+1})$$
$$= \mathbb{P}_{\widehat{M}}^{\pi_e}(o_h = i \mid s_{j+1} = k, \tau_j).$$

Therefore $\|\mathbf{P}_{\tau_j}\|_1 = 1$. Since $\|\widehat{\mathbb{O}}_{j+1}^{\dagger,w}\|_1 \le C_{\mathcal{O}}$, we further upper bound Eq. (7) as

$$\left\| \sum_{j=1}^{h-1} \sum_{\tau_j} \pi_e(\tau_j) \mathbf{P}_{\tau_j} \widehat{\mathbb{O}}_{j+1}^{\dagger,w} (\widehat{\mathbf{B}}_j - \mathbf{B}_j) \mathbf{b}(\tau_{j-1}) \right\|_1$$
$$\le C_{\mathcal{O}} \sum_{j=1}^{h-1} \sum_{\tau_j} \pi_e(\tau_j) \left\| (\widehat{\mathbf{B}}_j - \mathbf{B}_j) \mathbf{b}(\tau_{j-1}) \right\|_1.$$

Taking summation over $h$ from 1 to $H$, we obtain

$$\left| J(\pi_e) - J_{\widehat{M}}(\pi_e) \right| \le H C_{\mathcal{O}} \varepsilon_{\text{MLE}} + H C_{\mathcal{O}} \sum_{j=1}^{H-1} \sum_{\tau_j} \pi_e(\tau_j) \left\| (\widehat{\mathbf{B}}_j - \mathbf{B}_j) \mathbf{b}(\tau_{j-1}) \right\|_1. \tag{8}$$

From Eq. (5), we know that $\sum_{\tau_h} \pi_b(\tau_h) \times \left\| \left( \widehat{\mathbf{B}}_h - \mathbf{B}_h \right) \mathbf{b}(\tau_{h-1}) \right\|_1 \le 2 C_{\mathcal{O}} \varepsilon_{\text{MLE}}$. Therefore, we consider the ratio between estimation error of $\pi_e$ and estimation error of $\pi_b$ for step $h \in [H-1]$.

$$\frac{\sum_{\tau_h} \pi_e(\tau_h) \times \left\| (\widehat{\mathbf{B}}_h - \mathbf{B}_h) \mathbf{b}(\tau_{h-1}) \right\|_1}{\sum_{\tau_h} \pi_b(\tau_h) \times \left\| \left( \widehat{\mathbf{B}}_h - \mathbf{B}_h \right) \mathbf{b}(\tau_{h-1}) \right\|_1}$$
$$= \frac{\sum_{\tau_h} \pi_e(\tau_h) \mathbb{P}_{M^*}(\tau_{h-1}) \times \left\| (\widehat{\mathbf{B}}_h - \mathbf{B}_h) \mathbb{O}_h \mathbf{b}_{\mathcal{S}}(\tau_{h-1}) \right\|_1}{\sum_{\tau_h} \pi_b(\tau_h) \mathbb{P}_{M^*}(\tau_{h-1}) \times \left\| (\widehat{\mathbf{B}}_h - \mathbf{B}_h) \mathbb{O}_h \mathbf{b}_{\mathcal{S}}(\tau_{h-1}) \right\|_1}$$
$$= \frac{\sum_{o_h, a_h} \mathbb{E}_{\pi_e} \left[ \pi_e(a_h \mid \tau_{h-1}, o_h) \| (\widehat{\mathbf{B}}_h - \mathbf{B}_h) \mathbb{O}_h \mathbf{b}_{\mathcal{S}}(\tau_{h-1}) \|_1 \right]}{\sum_{o_h, a_h} \mathbb{E}_{\pi_b} \left[ \pi_b(a_h \mid \tau_{h-1}, o_h) \| (\widehat{\mathbf{B}}_h - \mathbf{B}_h) \mathbb{O}_h \mathbf{b}_{\mathcal{S}}(\tau_{h-1}) \|_1 \right]}.$$

In the first step, we use the relation $\mathbf{b}(\tau_h) = \mathbb{P}_{M^*}(\tau_h) \mathbb{O}_{h+1} \mathbf{b}_{\mathcal{S}}(\tau_h)$. Combining it with Eq. (8), we have

$$\left| J(\pi_e) - J_{\widehat{M}}(\pi_e) \right| \le H C_{\mathcal{O}} \varepsilon_{\text{MLE}} + 2 H^2 C_{\mathcal{O}}^2 C_{\text{eff},1} \varepsilon_{\text{MLE}},$$

where

$$C_{\text{eff},1} := \max_{h \in [H-1]} \frac{\sum_{o_h, a_h} \mathbb{E}_{\pi_e} \left[ \pi_e(a_h \mid \tau_{h-1}, o_h) \| (\widehat{\mathbf{B}}_h - \mathbf{B}_h) \mathbb{O}_h \mathbf{b}_{\mathcal{S}}(\tau_{h-1}) \|_1 \right]}{\sum_{o_h, a_h} \mathbb{E}_{\pi_b} \left[ \pi_b(a_h \mid \tau_{h-1}, o_h) \| (\widehat{\mathbf{B}}_h - \mathbf{B}_h) \mathbb{O}_h \mathbf{b}_{\mathcal{S}}(\tau_{h-1}) \|_1 \right]}.$$

We have proved the first part of the theorem. Next, we need to upper bound $C_{\text{eff},1}$. Let $x_l(o_h, a_h) \in \mathbb{R}^{|\mathcal{S}_h|}$ be the $l$-th row of $\left[ (\widehat{\mathbf{B}}_h(o_h, a_h) - \mathbf{B}_h(o_h, a_h) \mathbb{O}_h \right]$, then we have

$$C_{\text{eff},1} \le \max_{h \in [H-1]} \max_{o_h, a_h, l} \frac{\mathbb{E}_{\pi_e} \left[ \pi_e(a_h \mid \tau_{h-1}, o_h) |x_l(o_h, a_h)^\top \mathbf{b}_{\mathcal{S}}(\tau_{h-1})| \right]}{\mathbb{E}_{\pi_b} \left[ \pi_b(a_h \mid \tau_{h-1}, o_h) |x_l(o_h, a_h)^\top \mathbf{b}_{\mathcal{S}}(\tau_{h-1})| \right]}$$
$$\le \max_{h \in [H-1]} \max_{o_h, a_h, l} C_{\mathcal{A}} \frac{\mathbb{E}_{\pi_e} |x_l(o_h, a_h)^\top \mathbf{b}_{\mathcal{S}}(\tau_{h-1})|}{\mathbb{E}_{\pi_b} |x_l(o_h, a_h)^\top \mathbf{b}_{\mathcal{S}}(\tau_{h-1})|}.$$

To lower bound the denominator, we have

$$\mathbb{E}_{\pi_b} \left| x_l(o_h, a_h)^\top \mathbf{b}_{\mathcal{S}}(\tau_{h-1}) \right| = \mathbb{E}_{\pi_b} \frac{\left( x_l(o_h, a_h)^\top \mathbf{b}_{\mathcal{S}}(\tau_{h-1}) \right)^2}{|x_l(o_h, a_h)^\top \mathbf{b}_{\mathcal{S}}(\tau_{h-1})|}$$

$$\geq \mathbb{E}_{\pi_b} \frac{\left(x_l(o_h, a_h)^\top \mathbf{b}_{\mathcal{S}}(\tau_{h-1})\right)^2}{\|x_l(o_h, a_h)\|_2 \|\mathbf{b}_{\mathcal{S}}(\tau_{h-1})\|_2}.$$

$$\geq \frac{\sigma_{\min}(\Sigma_{\mathcal{H},h}) \|x_l(o_h, a_h)\|_2^2}{\|x_l(o_h, a_h)\|_2 \|\mathbf{b}_{\mathcal{S}}(\tau_{h-1})\|_2}$$

$$\geq \sigma_{\min}(\Sigma_{\mathcal{H},h}) \|x_l(o_h, a_h)\|_2. \tag{9}$$

Here $\Sigma_{\mathcal{H},h} = \mathbb{E}_{\pi_b} \left[\mathbf{b}_{\mathcal{S}}(\tau_{h-1}) \mathbf{b}_{\mathcal{S}}(\tau_{h-1})^\top\right]$. The first inequality is from Cauchy-Schwarz inequality and the last inequality is because $\|\mathbf{b}_{\mathcal{S}}(\tau_{h-1})\|_2 \leq 1$. The numerator is upper bounded by

$$\left|\mathbb{E}_{\pi_e} \left[x_l(o_h, a_h)^\top \mathbf{b}_{\mathcal{S}}(\tau_{h-1})\right]\right| \leq \|x_l(o_h, a_h)\|_2 \|\mathbb{E}_{\pi_e} \mathbf{b}_{\mathcal{S}}(\tau_{h-1})\|_2. \tag{10}$$

Combining Eq. (9) and Eq. (10),

$$C_{\text{eff},1} \leq C_{\mathcal{A}} C_{\mathcal{H}}.$$

The proof is completed. $\qquad\square$

## D.2 PROOF OF THEOREM 5

As in Appendix D.1, we first show a lemma for the operators $\mathbf{B}$ in the multi-step outcome scenario.

**Lemma 11.** *For any index $0 \leq j < h \leq H-1$, trajectory $\tau_j \in \prod_{h'=1}^{j}(\mathcal{O}_{h'} \times \mathcal{A})$, vector $x \in \mathbb{R}^{|\mathcal{F}_{j+1}|}$, policy $\pi$, and operator $\mathbf{B}$ satisfying Assumption D, we have*

$$\sum_{\tau_{h:j+1}} \|\mathbf{B}_{h:j+1} x\|_1 \times \pi(\tau_{h:j+1} \mid \tau_j) \leq C_{\mathcal{F}} \|x\|_1.$$

*Proof.* From the definition of $\mathbf{B}_{h:j+1}$, we have

$$\mathbf{B}_{h:j+1} x = \mathbf{B}_{h:j+1} U_{\mathcal{F},j+1} U_{\mathcal{F},j+1}^{\dagger,w} x.$$

Then, for any standard basis $\mathbf{e}_i \in \mathbb{R}^S$, we obtain

$$\sum_{\tau_{h:j+1}} \|\mathbf{B}_{h:j+1} U_{\mathcal{F},j+1} \mathbf{e}_i\|_1 \times \pi(\tau_{h:j+1} \mid \tau_j)$$

$$= \sum_{\tau_{h:j+1}} \sum_{f_{h+1}} \mathbb{P}_M^{\pi_b}(f_{h+1}, o_{h:j+1} \mid s_{j+1} = i, a_{h:j+1}) \cdot \pi(\tau_{h:j+1} \mid \tau_j)$$

$$= \sum_{\tau_{h:j+1}} \sum_{f_{h+1}} \mathbb{P}_M^{a_{h:j+1} \sim \pi, a_{h+1:} \sim \pi_b}(f_{h+1}, \tau_{h:j+1} \mid \tau_j, s_{j+1} = i) = 1.$$

Therefore

$$\sum_{\tau_{h:j+1}} \|\mathbf{B}_{h:j+1} x\|_1 \times \pi(\tau_{h:j+1} \mid \tau_j)$$

$$\leq \sum_{\tau_{h:j+1}} \|\mathbf{B}_{h:j+1} U_{\mathcal{F},j+1}\|_1 \|U_{\mathcal{F},j+1}^{\dagger,w} x\|_1 \times \pi(\tau_{h:j+1} \mid \tau_j)$$

$$\leq C_{\mathcal{F}} \|x\|_1.$$

The last step is from Assumption D and $\|U_{\mathcal{F},j+1}^{\dagger,w}\|_1 \leq \|\Sigma_{\mathcal{F},j+1}^{-1}\|_1 \|U_{\mathcal{F},j+1}^\top Z_{j+1}^{-1}\|_1 = C_{\mathcal{F}}$. $\qquad\square$

Then, we prove Theorem 5.

*Proof.* We first observe that for any $h \in [H-1]$,

$$\sum_{\tau_h} \pi_b(\tau_h) \times \|\widehat{\mathbf{b}}(\tau_h) - \mathbf{b}(\tau_h)\|_1 = \sum_{\tau_h, f_{h+1}} \left|\mathbb{P}_{\widehat{M}}^{\pi_b}(\tau_h, f_{h+1}) - \mathbb{P}_{M^*}^{\pi_b}(\tau_h, f_{h+1})\right| \leq \varepsilon_{\text{MLE}}.$$

The inequality is from Eq. (4). Hence, using Lemma 11, for the estimation error of $\pi_b$ at step $h \in [H-1]$, we have

$$\sum_{\tau_h} \pi_b(\tau_h) \times \left\| \left( \widehat{\mathbf{B}}_h(o_h, a_h) - \mathbf{B}_h(o_h, a_h) \right) \mathbf{b}(\tau_{h-1}) \right\|_1$$

$$\leq \sum_{\tau_h} \pi_b(\tau_h) \times \left\| \widehat{\mathbf{B}}_h(o_h, a_h)\widehat{\mathbf{b}}(\tau_{h-1}) - \mathbf{B}_h(o_h, a_h)\mathbf{b}(\tau_{h-1}) \right\|_1$$

$$+ \sum_{\tau_h} \pi_b(\tau_h) \times \left\| \widehat{\mathbf{B}}_h(o_h, a_h) \left( \widehat{\mathbf{b}}(\tau_{h-1}) - \mathbf{b}(\tau_{h-1}) \right) \right\|_1$$

$$\leq 2C_{\mathcal{F}}\varepsilon_{\text{MLE}}. \tag{11}$$

Next, we consider the reward of $\pi_e$ at each step. For step $h$, we construct policy $\mu_h$, which follows $\pi_e$ until step $h-1$ and follows $\pi_b$ from step $h$. It is clear that $\mu_h$ has the same expected reward as $\pi_e$ at step $h$. Let $r_h(f_h)$ be the reward of $f_h$ at step $h$, we have

$$\left| J(\pi_e) - J_{\widehat{M}}(\pi_e) \right| = \sum_{h=1}^{H} \left| \sum_{f_h} (\mathbb{P}_{\widehat{M}}^{\mu_h}(f_h) - \mathbb{P}_{M^*}^{\mu_h}(f_h))r_h(f_h) \right|$$

$$\leq \sum_{h=1}^{H} \sum_{f_h} \left| \mathbf{e}_{f_h}^{\top} \left( \sum_{\tau_{h-1}} \left( \widehat{\mathbf{B}}_{h-1} \cdots \widehat{\mathbf{B}}_1 \widehat{\mathbf{b}}_0 - \mathbf{B}_{h-1} \cdots \mathbf{B}_1 \mathbf{b}_0 \right) \times \pi_e(\tau_{h-1}) \right) \right|$$

$$= \sum_{h=1}^{H} \left\| \sum_{\tau_{h-1}} \left( \widehat{\mathbf{B}}_{h-1} \cdots \widehat{\mathbf{B}}_1 \widehat{\mathbf{b}}_0 - \mathbf{B}_{h-1} \cdots \mathbf{B}_1 \mathbf{b}_0 \right) \times \pi_e(\tau_{h-1}) \right\|_1.$$

Similar to the derivation of Theorem 4, for the case $h = 1$, according to Eq. (4), we have $\sum_{f_1} \left| \mathbb{P}_{\widehat{M}}^{\pi_b}(f_1) - \mathbb{P}_{M^*}^{\pi_b}(f_1) \right| \leq \varepsilon_{\text{MLE}}$. For $2 \leq h \leq H$, we use the telescoping in Eq. (6) and obtain

$$\left\| \sum_{\tau_{h-1}} \left( \widehat{\mathbf{B}}_{h-1} \cdots \widehat{\mathbf{B}}_1 \widehat{\mathbf{b}}_0 - \mathbf{B}_{h-1} \cdots \mathbf{B}_1 \mathbf{b}_0 \right) \times \pi_e(\tau_{h-1}) \right\|_1.$$

$$\leq \left\| \sum_{\tau_{h-1}} \pi_e(\tau_{h-1})\widehat{\mathbf{B}}_{h-1:1}(\widehat{\mathbf{b}}_0 - \mathbf{b}_0) \right\|_1 + \left\| \sum_{\tau_{h-1}} \pi_e(\tau_{h-1}) \sum_{j=1}^{h-1} \widehat{\mathbf{B}}_{h-1:j+1}(\widehat{\mathbf{B}}_j - \mathbf{B}_j)\mathbf{b}(\tau_{j-1}) \right\|_1.$$

For the first term, we have

$$\left\| \sum_{\tau_{h-1}} \pi_e(\tau_{h-1})\widehat{\mathbf{B}}_{h-1:1}(\widehat{\mathbf{b}}_0 - \mathbf{b}_0) \right\|_1 \leq \sum_{\tau_{h-1}} \left\| \widehat{\mathbf{B}}_{h-1:1}(\widehat{\mathbf{b}}_0 - \mathbf{b}_0) \right\|_1 \times \pi_e(\tau_{h-1})$$

$$\leq C_{\mathcal{F}}\|\widehat{\mathbf{b}}_0 - \mathbf{b}_0\|_1 \qquad \text{(Lemma 11)}$$

$$\leq C_{\mathcal{F}}\varepsilon_{\text{MLE}}. \qquad \text{(Eq. (4))}$$

For the second term, we have

$$\left\| \sum_{\tau_{h-1}} \pi_e(\tau_{h-1}) \sum_{j=1}^{h-1} \widehat{\mathbf{B}}_{h-1:j+1}(\widehat{\mathbf{B}}_j - \mathbf{B}_j)\mathbf{b}(\tau_{j-1}) \right\|_1$$

$$= \left\| \sum_{j=1}^{h-1} \sum_{\tau_j} \pi_e(\tau_j) \sum_{\tau_{h-1:j+1}} \pi_e(\tau_{h-1:j+1} \mid \tau_j) \times \widehat{\mathbf{B}}_{h-1:j+1}(\widehat{\mathbf{B}}_j - \mathbf{B}_j)\mathbf{b}(\tau_{j-1}) \right\|_1$$

$$= \left\| \sum_{j=1}^{h-1} \sum_{\tau_j} \pi_e(\tau_j)\mathbf{P}_{\tau_j}\widehat{U}_{\mathcal{F},j+1}^{\dagger,w}(\widehat{\mathbf{B}}_j - \mathbf{B}_j)\mathbf{b}(\tau_{j-1}) \right\|_1, \tag{12}$$

where

$$\mathbf{P}_{\tau_j} = \sum_{\tau_{h-1:j+1}} \pi_e(\tau_{h-1:j+1} \mid \tau_j) \times \widehat{\mathbf{B}}_{h-1:j+1} \widehat{U}_{\mathcal{F},j+1}$$

and we observe that

$$[\mathbf{P}_{\tau_j}]_{ik} = \sum_{\tau_{h-1:j+1}} \pi_e(\tau_{h-1:j+1} \mid \tau_j) \mathbb{P}_{\widehat{M}}^{\pi_b}(f_h = i, o_{h-1:j+1} \mid s_{j+1} = k, a_{h:j+1})$$

$$= \mathbb{P}_{\widehat{M}}^{a_{h-1:j+1} \sim \pi_e, a_{h:} \sim \pi_b}(f_h = i \mid s_{j+1} = k, \tau_j).$$

Therefore $\|\mathbf{P}_{\tau_j}\|_1 = 1$. Since $\|\widehat{U}_{\mathcal{F},j+1}^{\dagger,w}\|_1 \le C_{\mathcal{F}}$, we further upper bound Eq. (12) as

$$\left\| \sum_{j=1}^{h-1} \sum_{\tau_j} \pi_e(\tau_j) \mathbf{P}_{\tau_j} \widehat{U}_{\mathcal{F},j+1}^{\dagger,w} (\widehat{\mathbf{B}}_j - \mathbf{B}_j) \mathbf{b}(\tau_{j-1}) \right\|_1$$

$$\le C_{\mathcal{F}} \sum_{j=1}^{h-1} \sum_{\tau_j} \pi_e(\tau_j) \left\| (\widehat{\mathbf{B}}_j - \mathbf{B}_j) \mathbf{b}(\tau_{j-1}) \right\|_1.$$

Taking summation over $h$ from 1 to $H$, we get

$$\left| J(\pi_e) - J_{\widehat{M}}(\pi_e) \right| \le H C_{\mathcal{O}} \varepsilon_{\text{MLE}} + H C_{\mathcal{F}} \sum_{j=1}^{H-1} \sum_{\tau_j} \pi_e(\tau_j) \left\| (\widehat{\mathbf{B}}_j - \mathbf{B}_j) \mathbf{b}(\tau_{j-1}) \right\|_1. \quad (13)$$

According to Eq. (11), we have $\sum_{\tau_h} \pi_b(\tau_h) \times \left\| \left( \widehat{\mathbf{B}}_h - \mathbf{B}_h \right) \mathbf{b}(\tau_{h-1}) \right\|_1 \le 2 C_{\mathcal{F}} \varepsilon_{\text{MLE}}$. Therefore, the ratio between estimation error of $\pi_e$ and estimation error of $\pi_b$ for step $h \in [H-1]$ is written as:

$$\frac{\sum_{\tau_h} \pi_e(\tau_h) \times \left\| (\widehat{\mathbf{B}}_h - \mathbf{B}_h) \mathbf{b}(\tau_{h-1}) \right\|_1}{\sum_{\tau_h} \pi_b(\tau_h) \times \left\| \left( \widehat{\mathbf{B}}_h - \mathbf{B}_h \right) \mathbf{b}(\tau_{h-1}) \right\|_1}$$

$$= \frac{\sum_{\tau_h} \pi_e(\tau_h) \mathbb{P}_{M^*}(\tau_{h-1}) \times \left\| (\widehat{\mathbf{B}}_h - \mathbf{B}_h) U_{\mathcal{F},h} \mathbf{b}_{\mathcal{S}}(\tau_{h-1}) \right\|_1}{\sum_{\tau_h} \pi_b(\tau_h) \mathbb{P}_{M^*}(\tau_{h-1}) \times \left\| (\widehat{\mathbf{B}}_h - \mathbf{B}_h) U_{\mathcal{F},h} \mathbf{b}_{\mathcal{S}}(\tau_{h-1}) \right\|_1}$$

$$= \frac{\sum_{o_h, a_h} \mathbb{E}_{\pi_e} \left[ \pi_e(a_h \mid \tau_{h-1}, o_h) \|(\widehat{\mathbf{B}}_h - \mathbf{B}_h) U_{\mathcal{F},h} \mathbf{b}_{\mathcal{S}}(\tau_{h-1})\|_1 \right]}{\sum_{o_h, a_h} \mathbb{E}_{\pi_b} \left[ \pi_b(a_h \mid \tau_{h-1}, o_h) \|(\widehat{\mathbf{B}}_h - \mathbf{B}_h) U_{\mathcal{F},h} \mathbf{b}_{\mathcal{S}}(\tau_{h-1})\|_1 \right]}.$$

In the first step, we use the relation $\mathbf{b}(\tau_h) = \mathbb{P}_{M^*}(\tau_h) U_{\mathcal{F},h+1} \mathbf{b}_{\mathcal{S}}(\tau_h)$. Combining it with Eq. (13), we have

$$\left| J(\pi_e) - J_{\widehat{M}}(\pi_e) \right| \le H C_{\mathcal{F}} \varepsilon_{\text{MLE}} + 2 H^2 C_{\mathcal{F}}^2 C_{\text{eff},m} \varepsilon_{\text{MLE}},$$

where

$$C_{\text{eff},m} := \max_{h \in [H-1]} \frac{\sum_{o_h, a_h} \mathbb{E}_{\pi_e} \left[ \pi_e(a_h \mid \tau_{h-1}, o_h) \|(\widehat{\mathbf{B}}_h - \mathbf{B}_h) U_{\mathcal{F},h} \mathbf{b}_{\mathcal{S}}(\tau_{h-1})\|_1 \right]}{\sum_{o_h, a_h} \mathbb{E}_{\pi_b} \left[ \pi_b(a_h \mid \tau_{h-1}, o_h) \|(\widehat{\mathbf{B}}_h - \mathbf{B}_h) U_{\mathcal{F},h} \mathbf{b}_{\mathcal{S}}(\tau_{h-1})\|_1 \right]}.$$

Similar to Theorem 4, we further have $C_{\text{eff},m} \le C_{\mathcal{A}} C_{\mathcal{H}}$, the proof is completed. □

## E  PROOFS FOR SECTION 5

*Proof of Theorem 8.* Recall that

$$\varepsilon_{\text{approx}} := \min_{M \in \mathcal{M}} \frac{1}{n} \sum_{i=1}^{n} \left( \log \mathbb{P}_{M^*}^{\pi}(\tau_H^{(i)}) - \log \mathbb{P}_M^{\pi}(\tau_H^{(i)}) \right).$$

By invoking Lemma 9, with probability at least $1 - \delta$, we have

$$\sum_{\tau \in \prod_{h=1}^{H}(\mathcal{O}_h \times \mathcal{A})} \left| \mathbb{P}_{\widehat{M}}^{\pi_b}(\tau) - \mathbb{P}_{M^\star}^{\pi_b}(\tau) \right| \leq \mathcal{O}\left( \sqrt{\frac{\log \frac{|\mathcal{M}|}{\delta}}{n}} + \varepsilon_{\text{approx}} \right) := \widetilde{\varepsilon}_{\text{MLE}}.$$

Therefore, for any $h \in [H - 1]$, the following inequality holds,

$$\sum_{\tau_h, f_{h+1}} \left| \mathbb{P}_{\widehat{M}}^{\pi_b}(\tau_h, f_{h+1}) - \mathbb{P}_{M^*}^{\pi_b}(\tau_h, f_{h+1}) \right| \leq \widetilde{\varepsilon}_{\text{MLE}}.$$

The remaining proof is similar to the proof of Theorem 5. Finally, we get

$$\left| J(\pi_e) - J_{\widehat{M}}(\pi_e) \right| \leq HC_{\mathcal{F}} \widetilde{\varepsilon}_{\text{MLE}} + 2H^2 C_{\mathcal{F}}^2 C_{\text{eff},m} \widetilde{\varepsilon}_{\text{MLE}}.$$

Substituting $\widetilde{\varepsilon}_{\text{MLE}}$ finishes the proof. □

## F    TIGHER BOUND FOR MEMORYLESS $\pi_e$

The paper mostly focuses on evaluation history-dependent target policies $\pi_e$ in the previous section. Here, we show that we can obtain a tighter coverage coefficient for memoryless $\pi_e$. We present the results for multi-step outcome revealing, with the single-step case being similar.

**Theorem 12.** *Under the same condition as Theorem 5, and assuming $\pi_e$ is memoryless, with probability at least $1 - \delta$, we have*

$$\left| J(\pi_e) - J_{\widehat{M}}(\pi_e) \right| \leq \mathcal{O}\left( H^2 C_{\mathcal{F}}^2 \widetilde{C}_{\text{eff},m} \sqrt{\frac{\log \frac{|\mathcal{M}|}{\delta}}{n}} \right).$$

*where*

$$\widetilde{C}_{\text{eff},m} := \max_{h \in [H-1]} \frac{\sum_{o_h, a_h} \sum_{l=1}^{|\mathcal{F}_{h+1}|} \left| \mathbb{E}_{\pi_e} \left[ \pi_e(a_h \mid \tau_{h-1}, o_h) \left[ (\widehat{\mathbf{B}}_h - \mathbf{B}_h) U_{\mathcal{F},h} \right]_{l,:} \mathbf{b}_{\mathcal{S}}(\tau_{h-1}) \right] \right|}{\sum_{o_h, a_h} \mathbb{E}_{\pi_b} \left[ \pi_b(a_h \mid \tau_{h-1}, o_h) \| (\widehat{\mathbf{B}}_h - \mathbf{B}_h) U_{\mathcal{F},h} \mathbf{b}_{\mathcal{S}}(\tau_{h-1}) \|_1 \right]}.$$

Note that the numerator of coverage coefficient $C_{\text{eff},m}$ in Theorem 5 can be written as:

$$\sum_{o_h, a_h} \sum_{l=1}^{|\mathcal{F}_{h+1}|} \mathbb{E}_{\pi_e} \left| \pi_e(a_h \mid \tau_{h-1}, o_h) \left[ (\widehat{\mathbf{B}}_h - \mathbf{B}_h) U_{\mathcal{F},h} \right]_{l,:} \mathbf{b}_{\mathcal{S}}(\tau_{h-1}) \right|.$$

Compared to $C_{\text{eff},m}$, the numerator in $\widetilde{C}_{\text{eff},m}$ places the absolute operator outside of the expectation, resulting in a tighter coverage measurement. Mathematically similar tighter coverage coefficients have also been discovered in offline linear MDPs or offline MDPs with general function approximation. We refer readers to Jiang & Xie (2024) for more details.

*Proof of Theorem 12.* The proof is the same as for Theorem 5 up to bounding Eq. (12). For memoryless $\pi_e$, we observe that $\mathbf{P}_{\tau_j}$ is the same across different $\tau_j$. Therefore, we upper bound Eq. (12) as

$$\left\| \sum_{j=1}^{h-1} \sum_{\tau_j} \pi_e(\tau_j) \mathbf{P}_{\tau_j} \widehat{U}_{\mathcal{F},j+1}^{\dagger,w} (\widehat{\mathbf{B}}_j - \mathbf{B}_j) \mathbf{b}(\tau_{j-1}) \right\|_1$$

$$\leq C_{\mathcal{F}} \sum_{j=1}^{h-1} \left\| \sum_{\tau_j} \pi_e(\tau_j) (\widehat{\mathbf{B}}_j - \mathbf{B}_j) \mathbf{b}(\tau_{j-1}) \right\|_1.$$

Taking summation over $h$ from 1 to $H$, we get

$$\left| J(\pi_e) - J_{\widehat{M}}(\pi_e) \right| \leq HC_{\mathcal{F}} \varepsilon_{\text{MLE}} + HC_{\mathcal{F}} \sum_{j=1}^{H-1} \left\| \sum_{\tau_j} \pi_e(\tau_j) (\widehat{\mathbf{B}}_j - \mathbf{B}_j) \mathbf{b}(\tau_{j-1}) \right\|_1. \tag{14}$$

As before, we consider the ratio between estimation error of $\pi_e$ and estimation error of $\pi_b$ for step $h \in [H-1]$:

$$\frac{\left\|\sum_{\tau_h} \pi_e(\tau_h) \times (\widehat{\mathbf{B}}_h - \mathbf{B}_h)\mathbf{b}(\tau_{h-1})\right\|_1}{\sum_{\tau_h} \pi_b(\tau_h) \times \left\|\left(\widehat{\mathbf{B}}_h - \mathbf{B}_h\right)\mathbf{b}(\tau_{h-1})\right\|_1}$$

$$= \frac{\left\|\sum_{\tau_h} \pi_e(\tau_h)\mathbb{P}_{M^*}(\tau_{h-1}) \times (\widehat{\mathbf{B}}_h - \mathbf{B}_h)U_{\mathcal{F},h}\mathbf{b}_{\mathcal{S}}(\tau_{h-1})\right\|_1}{\sum_{\tau_h} \pi_b(\tau_h)\mathbb{P}_{M^*}(\tau_{h-1}) \times \left\|(\widehat{\mathbf{B}}_h - \mathbf{B}_h)U_{\mathcal{F},h}\mathbf{b}_{\mathcal{S}}(\tau_{h-1})\right\|_1}$$

$$= \frac{\sum_{o_h,a_h} \sum_{l=1}^{|\mathcal{F}_{h+1}|} \left|\mathbb{E}_{\pi_e}\left[\pi_e(a_h \mid \tau_{h-1}, o_h)\left[(\widehat{\mathbf{B}}_h - \mathbf{B}_h)U_{\mathcal{F},h}\right]_{l,:}\mathbf{b}_{\mathcal{S}}(\tau_{h-1})\right]\right|}{\sum_{o_h,a_h}\mathbb{E}_{\pi_b}\left[\pi_b(a_h \mid \tau_{h-1}, o_h)\|(\widehat{\mathbf{B}}_h - \mathbf{B}_h)U_{\mathcal{F},h}\mathbf{b}_{\mathcal{S}}(\tau_{h-1})\|_1\right]}.$$

Combining it with Eq. (14), we have

$$\left|J(\pi_e) - J_{\widehat{M}}(\pi_e)\right| \leq HC_{\mathcal{F}}\varepsilon_{\text{MLE}} + 2H^2 C_{\mathcal{F}}^2 \widetilde{C}_{\text{eff},m}\varepsilon_{\text{MLE}},$$

where

$$\widetilde{C}_{\text{eff},m} := \max_{h \in [H-1]} \frac{\sum_{o_h,a_h} \sum_{l=1}^{|\mathcal{F}_{h+1}|} \left|\mathbb{E}_{\pi_e}\left[\pi_e(a_h \mid \tau_{h-1}, o_h)\left[(\widehat{\mathbf{B}}_h - \mathbf{B}_h)U_{\mathcal{F},h}\right]_{l,:}\mathbf{b}_{\mathcal{S}}(\tau_{h-1})\right]\right|}{\sum_{o_h,a_h}\mathbb{E}_{\pi_b}\left[\pi_b(a_h \mid \tau_{h-1}, o_h)\|(\widehat{\mathbf{B}}_h - \mathbf{B}_h)U_{\mathcal{F},h}\mathbf{b}_{\mathcal{S}}(\tau_{h-1})\|_1\right]}.$$

The proof is completed. $\qquad\square$

## G  COMPARISON BETWEEN SINGLE-STEP AND MULTI-STEP OUTCOME REVEALING

In the main text, we claim that multi-step outcome revealing (Assumption D) is a more lenient assumption than its single-step counterpart (Assumption E), and provide intuitions based on the confusion-matrix interpretation of $\Sigma_{\mathcal{F},h}$ and $\Sigma_{\mathcal{O},h}$. However, a rigorous quantitative argument is missing, which we discuss in this section. Ideally, what we want to show is that $C_{\mathcal{F}} \leq C_{\mathcal{O}}$ (assuming both upper bounds are tight); if so, bounded $C_{\mathcal{O}}$ (Assumption E) would immediately imply the same bound on $C_{\mathcal{F}}$ (Assumption D), showing that the latter is a weaker assumption.

Below we first show in Appendix G.1 that this can be proved up to a factor of $|\mathcal{S}_h|$ when $\pi_b$ is memoryless; the additional factor is due to the use of matrix 1-norm. For the more general case where $\pi_b$ can be history-dependent, the analysis for memoryless $\pi_b$ breaks down, which reveals some unnaturalness in the way we define $\Sigma_{\mathcal{F},h}$ and $\Sigma_{\mathcal{O},h}$ (which are inherited from prior works). We argue in Appendix G.2 that we can re-define $\Sigma_{\mathcal{F},h}$ and $\Sigma_{\mathcal{O},h}$ in a more natural manner that accounts for the latent state distribution under $\pi_b$, which will allow for a quantitative comparison between $\Sigma_{\mathcal{F},h}$ and $\Sigma_{\mathcal{O},h}$; furthermore, all the results in the main paper hold up to minor changes under the new definitions.

### G.1  MEMORYLESS $\pi_b$

We first compare $C_{\mathcal{F}}$ and $C_{\mathcal{O}}$ quantitatively assuming memoryless $\pi_b$. We can express $\Sigma_{\mathcal{F},h}$ as:

$$\Sigma_{\mathcal{F},h} = |\mathcal{S}_h| \sum_{f_h} \bar{z}(f_h)\bar{\mathbf{u}}(f_h)\bar{\mathbf{u}}(f_h)^\top,$$

where $\bar{z}(f_h) = \frac{z(f_h)}{|\mathcal{S}_h|}$ is the probability of observing $f_h$ when $s_h$ is uniformly distributed. Moreover, define $\text{Pr}'_{\pi_b}(\cdot)$ as a joint distribution over $s_h$ and $f_h$, where $s_h$ is uniformly sampled, and $f_h$ is rolled out from $s_h$ using $\pi_b$ (note that this distribution is only well-defined for memoryless $\pi_b$), and we have

$$[\bar{\mathbf{u}}(f_h)]_i = \frac{[\mathbf{u}(f_h)]_i}{z(f_h)} = \text{Pr}'_{\pi_b}(s_h = i \mid f_h).$$

Here, $\bar{\mathbf{u}}$ serves as an inverse belief state vector, predicting the current state based on future trajectory rather than the history. Similarly, $\Sigma_{\mathcal{O},h}$ can be written as:

$$\Sigma_{\mathcal{O},h} = |\mathcal{S}_h| \sum_{o_h} \bar{w}(o_h)\bar{\mathbf{u}}(o_h)\bar{\mathbf{u}}(o_h)^\top,$$

where $\bar{w}(o_h) = \frac{w(o_h)}{|\mathcal{S}_h|}$ represents the probability of observing $o_h$ when $s_h$ is uniformly distributed and $[\bar{\mathbf{u}}(o_h)]_i = \mathrm{Pr}'_{\pi_b}(s_h = i \mid o_h)$. (Note that $\bar{\mathbf{u}}(o_h)$ has no dependence on $\pi_b$ since variables after $a_h$ are not involved, but the distribution of variables is consistent with $\mathrm{Pr}'_{\pi_b}$.) Next, we show that $\|\Sigma_{\mathcal{F},h}\|_2 \geq \|\Sigma_{\mathcal{O},h}\|_2$. For any vector $\mathbf{a} \in \mathbb{R}^{|\mathcal{S}_h|}$, we have

$$\begin{aligned}
\mathbf{a}^\top \Sigma_{\mathcal{O},h}\mathbf{a} &= |\mathcal{S}_h|\mathbb{E}_{o_h}\left[(\mathbf{a}^\top\bar{\mathbf{u}}(o_h))^2\right] \\
&= |\mathcal{S}_h|\mathbb{E}_{o_h}\left[\left(\mathbb{E}_{f_h|o_h}[\mathbf{a}^\top\bar{\mathbf{u}}(f_h)]\right)^2\right] \\
&\leq |\mathcal{S}_h|\mathbb{E}_{o_h}\mathbb{E}_{f_h|o_h}\left[\left(\mathbf{a}^\top\bar{\mathbf{u}}(f_h)\right)^2\right] \\
&= \mathbf{a}^\top\Sigma_{\mathcal{F},h}\mathbf{a}.
\end{aligned}$$

The inequality is from Jensen's inequality. Therefore, we have:

$$\|\Sigma_{\mathcal{F},h}^{-1}\|_1 \leq \sqrt{|\mathcal{S}_h|}\|\Sigma_{\mathcal{F},h}^{-1}\|_2 \leq \sqrt{|\mathcal{S}_h|}\|\Sigma_{\mathcal{O},h}^{-1}\|_2 \leq |\mathcal{S}_h|\|\Sigma_{\mathcal{O},h}^{-1}\|_1.$$

This implies whenever single-step outcome revealing assumption is satisfied with bound $C_{\mathcal{O}}$, the multi-step revealing assumption also holds with $C_{\mathcal{F}} \leq \max_h |\mathcal{S}_h|C_{\mathcal{O}}$.

## G.2   HISTORY-DEPENDENT $\pi_b$

Unfortunately, the above analysis only works for memoryless $\pi_b$, and breaks down if $\pi_b$ is history-dependent: a key step was to interpret $\bar{\mathbf{u}}(f_h)$ as $[\bar{\mathbf{u}}(f_h)]_i = \mathrm{Pr}'_{\pi_b}(s_h = i \mid f_h)$, which is only possible when $\pi_b$ is memoryless. The root problem is that we want to take $[\mathbf{u}(f_h)]_i = \mathbb{P}^{\pi_b}(f_h \mid s_h = i)$ and use Bayes rule to convert it to the posterior over $s_h$ given $f_h$, so that we can interpret $\Sigma_{\mathcal{F},h}$ as the confusion matrix of predicting $s_h$ from $f_h$. Since the distribution is under $\pi_b$, it implicitly defines $d_h^{\pi_b}$ as the "label prior" for $s_h$, but our previous definitions enforce an unnatural and arbitrary uniform prior over $s_h$ (which stems from the $\mathbf{1}_{\mathcal{S}_h}$ in $Z_h = \mathrm{diag}(U_{\mathcal{F},h}\mathbf{1}_{\mathcal{S}_h})$).

To fix this, we show that we can re-define $\Sigma_{\mathcal{F},h}$ and $\Sigma_{\mathcal{O},h}$ in a way that is more natural and respects the $d_h^{\pi_b}$ prior for latent states, and all results in the main text hold up to minor changes, as will be explained. Concretely, let $\mathbf{p}_h = d_h^{\pi_b}$ for concision. We introduce a new weight matrix $Z_h^{\mathbf{P}_h} := \mathrm{diag}(U_{\mathcal{F},h}\mathbf{p}_h)$. Under this formulation, we have $Z_h^{\mathbf{P}_h}(f_h) = \mathrm{Pr}_{\pi_b}(f_h)$, which corresponds to the marginal probability of $f_h$ under $\pi_b$. With this adjustment, we propose the following new assumption.

**Assumption F** (Multi-Step Outcome Revealing with Memory-Based Behavior Policies). Define

$$\Sigma_{\mathcal{F},h}^{\mathbf{P}_h} := \mathrm{diag}(\mathbf{p}_h)U_{\mathcal{F},h}^\top(Z_h^{\mathbf{P}_h})^{-1}U_{\mathcal{F},h}.$$

We assume that $\|(\Sigma_{\mathcal{F},h}^{\mathbf{P}_h})^{-1}\|_1 \leq \widetilde{C}_{\mathcal{F}}, \forall h \in [H-1]$ for some $\widetilde{C}_{\mathcal{F}} < \infty$.

Here, $\Sigma_{\mathcal{F},h}^{\mathbf{P}_h}$ can be interpreted as the confusion matrix of latent states $s_h$ with respect to the following process: we first sample $f_h$ according to its marginal probability under $\pi_b$, $Z_h^{\mathbf{P}_h}(f_h) = \mathrm{Pr}_{\pi_b}(f_h)$. Then, conditioned on $f_h$, we independently sample two latent states, $s_h$ and $s'_h$, both from $\mathrm{Pr}_{\pi_b}(\cdot \mid f_h)$. Note that the joint distribution of $(s_h, f_h)$ in this process is consistent with that under $\pi_b$, so we abuse the notation $\mathrm{Pr}_{\pi_b}(\cdot)$ to refer to this distribution (which is augmented with a $s'_h$ variable). The $(i,j)$-th entry of $\Sigma_{\mathcal{F},h}^{\mathbf{P}_h}$ corresponds to the probability $\mathrm{Pr}_{\pi_b}(s'_h = i \mid s_h = j)$ since

$$[\Sigma_{\mathcal{F},h}^{\mathbf{P}_h}]_{ij} = \sum_k \mathrm{Pr}_{\pi_b}(s'_h = i \mid f_h = k)\mathrm{Pr}_{\pi_b}(f_h = k \mid s_h = j) = \mathrm{Pr}_{\pi_b}(s'_h = i \mid s_h = j).$$

This holds due to the conditional independence between $s_h$ and $s'_h$ given $f_h$, as defined by the sampling process. Similar to the properties of $\Sigma_{\mathcal{F},h}$, when $f_h$ deterministically predicts $s_h$, we have $\Sigma_{\mathcal{F},h}^{\mathbf{P}_h} = \mathbf{I}$. This interpretation justifies the boundedness of $\|(\Sigma_{\mathcal{F},h}^{\mathbf{P}_h})^{-1}\|_1$ as an assumption. In fact, when $\mathbf{p}_h$ is uniform, we have $\Sigma_{\mathcal{F},h}^{\mathbf{P}_h} = \Sigma_{\mathcal{F},h}$; when $\mathbf{p}_h$ is not uniform, $\Sigma_{\mathcal{F},h}^{\mathbf{P}_h}$ is more natural as it does not artificially and arbitrarily inject the uniform prior into the definition.

**Comparison between Single-step and Multi-step Revealing**    For the single-step outcome revealing assumption, we can similarly replace $\Sigma_{\mathcal{O},h}$ with $\Sigma_{\mathcal{O},h}^{\mathbf{P}_h} := \mathrm{diag}(\mathbf{p}_h)\mathbb{O}_h^\top (W_h^{\mathbf{P}_h})^{-1}\mathbb{O}_h$, where $W_h^{\mathbf{P}_h} := \mathrm{diag}(\mathbb{O}_h\mathbf{p}_h)$. The comparison between $\|(\Sigma_{\mathcal{F},h}^{\mathbf{P}_h})^{-1}\|_1$ and $\|(\Sigma_{\mathcal{O},h}^{\mathbf{P}_h})^{-1}\|_1$ follows a similar analysis as Appendix G.1, where $\mathrm{Pr}'_{\pi_b}(\cdot)$ will be replaced with the more natural $\mathrm{Pr}_{\pi_b}(\cdot)$.

**Changes to the Main Analyses**    We now show that the main results of our work are mostly unaffected if we switch to the new assumptions. Taking the upper-bound analysis (e.g., Theorem 5) as an example, $C_{\mathcal{F}}$ enters the analysis through bounding the norm of the OOM parameterization of the model (see e.g., Lemma 11). With the new definition of $\Sigma_{\mathcal{F},h}^{\mathbf{P}_h}$, all we need is to change the weighted pseudo-inverse used in the OOM representation accordingly: let

$$U_{\mathcal{F},h}^{\dagger,\mathbf{P}_h} = (\Sigma_{\mathcal{F},h}^{\mathbf{P}_h})^{-1}\mathrm{diag}(\mathbf{p}_h)U_{\mathcal{F},h}^\top (Z_h^{\mathbf{P}_h})^{-1},$$

which leads to the new operator:

$$\widetilde{\mathbf{B}}_h(o,a) = U_{\mathcal{F},h+1}\mathbb{T}_{h,a}\mathrm{diag}(\mathbb{O}_h(o\mid\cdot))U_{\mathcal{F},h}^{\dagger,\mathbf{P}_h}.$$

Since

$$\|U_{\mathcal{F},h}^{\dagger,\mathbf{P}_h}\|_1 \le \|(\Sigma_{\mathcal{F},h}^{\mathbf{P}_h})^{-1}\|_1\|\mathrm{diag}(\mathbf{p}_h)U_{\mathcal{F},h}^\top (Z_h^{\mathbf{P}_h})^{-1}\|_1 \le \widetilde{C}_{\mathcal{F}},$$

one can verify that Lemma 11 still holds for $\widetilde{\mathbf{B}}_h$ with $C_{\mathcal{F}}$ replaced by $\widetilde{C}_{\mathcal{F}}$. The rest of the analysis follows the same as in Appendix D.

**Future-dependent Value Function Construction**    Besides our analyses, changes in the definitions of $C_{\mathcal{F}}$ will also affect the comparison with the model-free results from Zhang & Jiang (2024) as cited in the first row of Table 1. As it turns out, these results also hold under the new definition. In Uehara et al. (2023a); Zhang & Jiang (2024), $C_{\mathcal{F}}$ is involved in their analysis when they use $\Sigma_{\mathcal{F},h}$ to construct the future-dependent value function (FDVF) (Uehara et al., 2023a) and bound it range. Here we show that we can also use the newly defined $\Sigma_{\mathcal{F},h}^{\mathbf{P}_h}$ to construct FDVF, whose range depends on $\widetilde{C}_{\mathcal{F}}$, and the rest of their analyses still hold.

A FDVF $V_{\mathcal{F}}$ satisfies that $\forall h$,

$$U_{\mathcal{F},h}^\top \times V_{\mathcal{F},h} = V_{\mathcal{S},h}^{\pi_e},$$

where $V_{\mathcal{S},h}^{\pi_e}$ is the latent-state value function of the memoryless evaluation policy $\pi_e$, i.e., the expected sum of rewards from step $h$ onwards conditioned on a latent state. Let $R^+(f_h) := \sum_{h'=h}^H R(o_{h'})$ be the Monte-Carlo return in $f_h$ and $Z_h^{R,\mathbf{P}_h}(f_h) := Z_h^{\mathbf{P}_h}(f_h)/R^+(f_h)$. We construct the FDVF as:

$$V_{\mathcal{F},h} = (Z_h^{R,\mathbf{P}_h})^{-1}U_{\mathcal{F},h}\mathrm{diag}(\mathbf{p}_h)(\Sigma_{\mathcal{F},h}^{R,\mathbf{P}_h})^{-\top}V_{\mathcal{S},h}^{\pi_e}, \quad \text{where } \Sigma_{\mathcal{F},h}^{R,\mathbf{P}_h} := \mathrm{diag}(\mathbf{p}_h)U_{\mathcal{F},h}^\top (Z_h^{R,\mathbf{P}_h})^{-1}U_{\mathcal{F},h}.$$

We can then make a similar outcome coverage assumption as in Assumption 9 of Zhang & Jiang (2024), ensuring the boundedness of $V_{\mathcal{F},h}$. For the on-policy case $\pi_b = \pi_e$, one can verify that the constructed $V_{\mathcal{F}}$ exactly recovers $R^+$, which is naturally bounded by $H$.

