# OpenReview forum: "Statistical Tractability of Off-policy Evaluation of History-dependent Policies in POMDPs"
_ICLR.cc/2025/Conference — ICLR 2025 Poster_

### Official Review · Reviewer_8ySD · 2024-11-02

**Soundness:** 3
**Presentation:** 3
**Contribution:** 3
**Rating:** 6
**Confidence:** 2

**Summary:**

This paper investigates off-policy evaluation in POMDPs. Compared to previous work, this paper considers more general target policies that depend on the entire observable history. The authors first provide hardness results showing that a model-free algorithm can not output a near-optimal evaluation with polynomial sample complexity. To handle this challenge, the authors adopt a model-based method based on MLE. The authors further prove that under certain assumptions, even considering more general target policies, the proposed model-based method can output near-optimal evaluation with polynomial sample complexity. Finally, the authors weaken the realization assumption under misspecification cases.

**Strengths:**

(1) The paper provides hardness results for model-free methods, which motivates the authors to adopt the model-based methods.

(2) This paper weakens several previous assumptions on target policies and POMDPs, and proves the polynomial sample complexity upper bound for model-based methods.

(3) The paper is well-organized and easy to follow.

**Weaknesses:**

It seems that the **Pre-filtering** and **MLE** parts of algorithms are not computationally efficient. **Pre-filtering** requires to verify each of the models in set $\mathcal{M}$. Is there any efficient way to implement these two steps?

**Questions:**

Should the policy $\pi$ in Eq. (1) be behavior policy $\pi_b$?

 I wonder if assumption A can be relaxed, for example, estimated via the offline data set? And how would this relax affect the MLE in Algorithm?

---

> ### Author Response · Authors · 2024-11-15
>
> We thank the reviewer for their valuable comments and respond to the weaknesses and questions below.
>
> ---
>
> **Pre-filtering is computationally inefficient**
>
> You are right. However, we focus on the statistical tractability in this work, and the same pre-filtering step has also appeared in MLE algorithms for online learning of POMDPs (Liu et al. 2022a). In fact, even without pre-filtering, MLE itself is generally computationally inefficient for latent-variable models like POMDPs, and standard approximations such as Expectation-Maximization (EM) do not have strong theoretical guarantees. That said, the pre-filtering step may inspire practical regularization methods (e.g., using $C\_\mathcal{F, h}$ as a regularizer to bias learning towards models with smaller $C\_\mathcal{F, h}$) to improve the performance of heuristic approximations of MLE, but that is beyond the scope of this paper as our focus is on the fundamental theoretical understanding.
>
> ---
>
> **$\pi$ in Eq.(1)**
>
> You are right that it should be $\pi\_b$; we will correct. In fact the policy does not matter here: $\log P\_{M}^{\pi}(\tau)$ can be broken into log of transition probabilities and log of action probabilities (under $\pi$), and the latter has no dependence on the model $M$ and thus has 0 gradient. We wrote in this form because it makes the application of MLE guarantees obvious, but we will add additional details to clarify.
>
> ---
>
> **Relaxing Assumption A & whether it can be estimated**
>
> We guess you mean whether we can handle unknown $\pi\_b$ and estimate it from data; we believe so. Note that MLE itself does not require knowledge of $\pi\_b$ (as explained above). The only step in the algorithm that requires $\pi\_b$ is actually pre-filtering in the multi-step revealing setting: to compute $\Sigma\_{\mathcal{F}, h}$ we need to know $U\_{\mathcal{F}, h}$, which is the probability of future sequences given different hidden state (under a candidate model dynamics). The probability of actions need to be given by $\pi\_b$. When $\pi\_b$ is unknown, the standard treatment is to assume a realizable policy class and learn it from data using behavior cloning. It remains to show that the calculation of $\Sigma\_{\mathcal{F}, h}$ is robust to the estimation error of $\pi\_b$ through behavior cloning, which we believe is possible.
>
> Also, a main message of our work is to compare model-free and model-based algorithms, and the model-free algorithms of Uehara et al. (2023a) and Zhang & Jiang (2024) also require knowledge of $\pi\_b$, so simply assuming a known $\pi\_b$ allows for the cleanest comparison.

---

### Official Review · Reviewer_CtM3 · 2024-11-03

**Soundness:** 3
**Presentation:** 3
**Contribution:** 3
**Rating:** 6
**Confidence:** 4

**Summary:**

This paper focus on OPE problems in unconfounded POMDPs with large observation spaces and history-dependent target policies. While previous work addressed the problem for memoryless target policies under certain coverage assumptions, the authors extended these results to history-dependent target policies and prove polynomial complexity.

**Strengths:**

1. The paper addresses an important gap in the literature by extending OPE to history-dependent target policies in POMDPs.
2. The authors provide solid theoretical results, including both hardness proofs and positive guarantees for model-based algorithm.
3. The paper is well-organized, with clear definitions, assumptions, and theorem statements that make the paper accessible.

**Weaknesses:**

1. The paper does not include experimental results to validate the theoretical findings or to demonstrate the practical effectiveness of the MLE-based algorithm.
2. The model-based algorithm is not robust to state-space mismatch or model misspecification.

**Questions:**

1. Can authors provide any empirical or simulation results by the proposed algorithm on the setting described in the paper?
2. Are there any possibilities that can handle the state-space mismatch problem specified in Theorem 7?

---

> ### Author Response · Authors · 2024-11-15
>
> We thank the reviewer for their valuable comments and respond to the weaknesses and questions below.
>
> ---
>
> **Empirical results**
>
> Our work aims to provide theoretical understanding of fundamental limits of OPE in POMDPs. A major bottleneck in experiments is that the MLE model-based algorithm is known to be computationally inefficient in latent-variable models such as POMDPs, and popular approximations based on Expectation-Maximization (EM) do not have strong theoretical guarantees. Nevertheless, we believe that the theoretical separation between model-free and model-based methods in this setting is interesting in its own right. Note that similar separation results in other areas of the RL literature are mostly pure theory, too (see e.g., Sun et al. 2019).
>
> ---
>
> **State-space mismatch**
>
> Contrary to what the reviewer wrote, state-space mismatch is actually handled in Theorem 8, which is a positive result that complements the negative result in Theorem 7. The problem is not about the algorithm, but how the assumptions are stated and extended (from the realizable case to the state-space mismatch case). If we take the earlier assumptions (especially Assumption D) at face value, Theorem 7 tells us that the MLE algorithm fails, but that is due to us misunderstanding what Assumption D really means and which models it is really imposed on ($M^*$ vs. models in $\mathcal{M}$). Theorem 8 shows that once we impose Assumption D on models in $\mathcal{M}$, which is an alternative interpretation of imposing it on $M^*$ + pre-filtering in the realizable setting, the polynomial guarantee can be recovered. The contrast between Theorems 7 and 8 teach us something interesting about the fundamental nature of the revealing assumptions, which is only exposed under state-space mismatch and rarely touched by prior works.

---

### Official Review · Reviewer_2cH2 · 2024-11-04

**Soundness:** 3
**Presentation:** 3
**Contribution:** 3
**Rating:** 6
**Confidence:** 3

**Summary:**

This paper explores the statistical tractability of off-policy evaluation (OPE) for history-dependent policies in unconfounded partially observable Markov decision processes (POMDPs) with large observation spaces. The authors first establish information-theoretic hardness results for model-free algorithms when evaluating history-dependent policies, highlighting the increased complexity compared to memoryless policies. To address these challenges, they then propose a model-based approach that enables efficient OPE for history-dependent policies.

**Strengths:**

The findings in this paper are interesting. It demonstrates that no sample-efficient model-free OPE algorithm exists for POMDPs when evaluating history-dependent policies, whereas a model-based approach can achieve polynomial sample complexity. This highlights the effectiveness of model-based methods in POMDPs, and perhaps inspires future research on model-based RL for POMDPs. The theoretical analysis is solid and strong, and the writing is logically structured.

**Weaknesses:**

1. The writing could be improved to be more accessible to readers who may not be deeply familiar with this topic. For example, in the introduction, the authors could provide additional insights into belief and outcome coverage when these concepts are first introduced.

2. Another area for improvement in the writing would be providing a broader perspective on the paper’s contributions. Since the main contribution is extending existing results from memoryless policies to history-dependent policies, it would be helpful to include a high-level discussion in the introduction on (1) why history-dependent policies are particularly important in POMDP settings, and (2) what major challenges arise when attempting to apply existing analysis techniques for memoryless policies to history-dependent policies.

**Questions:**

I have a concern regarding the MLE part. From my understanding, it seems that $\epsilon_{MLE}$ may implicitly depend on the horizon length $H$. Could you please clarify and elaborate on this point? If so, how would this dependence impact the main results in the paper?

---

> ### Author Response · Authors · 2024-11-15
>
> We thank the reviewer for their valuable comments and will improve the writing accordingly. Below we respond to the weaknesses and questions.
>
> ---
>
> **Importance of history-dependent policies**
>
> In POMDPs, optimal policies are generally history dependent. In particular, optimal policies can take the form of a mapping from belief states to actions, and the belief state itself is a function of history. Therefore, it is very natural to consider history-dependent policies in POMDPs, and the restriction to memoryless policies in prior works is due to technical difficulties. We can add more explanation on this point in revision.
>
> ---
>
> **Major challenges in applying existing analysis techniques for memoryless policies to history-dependent policies**
>
> The prior works of Uehara et al. (2023a) and Zhang & Jiang (2024) provide analyses of a model-free algorithm. As our Table 1 shows, **model-free algorithms provably cannot handle general history-dependent policies** under the assumptions of prior works (e.g., belief and outcome coverages). This is information-theoretic hardness (Theorem 3), so no amount of clever analyses can provide a rescue. Therefore, we need to consider stronger algorithms (model-based) to circumvent the hardness, which requires different analyses.
>
> ---
>
> **$H$-dependence on $\epsilon\_{MLE}$**
>
> **$\epsilon\_{MLE}$ does not depend on $H$ in any way.** Please refer to the equation above Eq.(3) on page 16, where an explicit expression for $\epsilon\_{MLE}$ is given and it does not depend on $H$. We are not sure why the reviewer makes this comment, but maybe you are worried that the $\epsilon\_{MLE}$ bounds the TV error of estimating the distribution of trajectories, which is supported on a very large space (the space of trajectories has a large cardinality that is exponential in horizon $H$). The key here is that the model class induces a hypothesis class of distributions over trajectories, and the learning algorithm is essentially performing density estimation using this hypothesis class. When the model class has limited capacity (this corresponds to the $\log|\mathcal{M}|$ term in $\epsilon\_{MLE}$), the TV error of density estimation will be bounded in a way that is **independent of the size of the support**. The size of the support is only incurred when you estimate without a hypothesis class (i.e., a "tabular" approach), using the empirical frequency that each trajectory appears in the data, but that is not the case here.

---

> > ### Comment · Reviewer_2cH2 · 2024-11-19
> >
> > Thank you for your detailed reply, and your willingness to add more explanation in revision.
> >
> > Regarding the $H$-dependence of $\epsilon_{\text{MLE}}$, I realize I might not have been entirely clear in my previous comments, but your guess about my concern is correct. I have reviewed Eq.(3) on page 16, and I would like to better understand how the size of $\mathcal{M}$ scales with $H$. Specifically, I am unsure about the precise definition of the model class $\mathcal{M}$. Could you provide a simple example to clarify this?
> >
> > Based on my current understanding, if the model class $\mathcal{M}$ represents the transition and reward models, then $|\mathcal{M}|$ would not scale with $H$. However, a polynomial $H$-term might appear as a coefficient in Eq.(3) because the estimation error should occur at least H times when we consider the marginal distribution of a whole trajectory. On the other hand, if $\mathcal{M}$ corresponds to the density of the entire trajectory, as your reply suggests, we can interpret this as a density estimation problem. In this case, while the polynomial $H$-term may not appear as a coefficient in Eq.(3), $|\mathcal{M}|$ should still depend on $H$.
> >
> > For instance, if we use a normal distribution for density estimation, the parameters would include the mean and covariance of the trajectory. Here, the mean would scale linearly with $H$, and the covariance would scale quadratically with $H$, reflecting the trajectory's dependence on $H$. If we assume that the model class $\mathcal{M}$ does not scale with $H$, it would imply that $\mathcal{M}$ is extremely limited, which could introduce significant bias.
> >
> > Could you confirm whether my understanding is correct or if I have missed something? Thank you for your clarification.

---

> > > ### Author Response · Authors · 2024-11-19
> > >
> > > Dear reviewer,
> > >
> > > $\mathcal{M}$ is the space of transition and reward models, but its size also bounds the size of the **(automatically) induced** density class. In particular, given realizability $M^\star \in \mathcal{M}$, we know that the distribution over trajectories induced by the behavior policy $\pi\_b$, i.e., our data distribution, satisfies $P_{M^\star}^{\pi\_b}(\cdot) \in \\\{P_{M}^{\pi\_b}(\cdot): M \in \mathcal{M}\\\}$, and this density class $\\\{P_{M}^{\pi\_b}(\cdot): M \in \mathcal{M}\\\}$ is what we run MLE over (see Eq.(1) on page 6, where $P\_M^\pi$ should be $P\_M^{\pi\_b}$ and it's a minor typo). Moreover, it should be clear that $|\\\{P_{M}^{\pi\_b}(\cdot): M \in \mathcal{M}\\\}| \le |\mathcal{M}|$.
> > >
> > > You tried imagining two cases (whether $\mathcal{M}$ is a class of transition and reward models or it's a density class), and since the answer is the former (the density class is automatically induced), we answer your question for this case alone. You asked:
> > >
> > > > However, a polynomial $H$-term might appear as a coefficient in Eq.(3) because the estimation error should occur at least H times when we consider the marginal distribution of a whole trajectory.
> > >
> > > We do not quite get what you mean. Note that $P_M^{\pi\_b}(\tau)$ is a distribution over the entire trajectory, so we run MLE only once, jointly over the entire trajectory (Eq.(1)), instead of $H$ times for "marginal distribution" as you suggested.
> > >
> > > Finally, we emphasize that the way the MLE component works in our paper is very standard and consistent with the algorithm and analyses in previous POMDP literature and model-based RL literature, such as Liu et al 2022(a) and the references therein. The main technique comes from the seminal paper of Tong Zhang [1], which was introduced into RL theory by the Flambe paper by Agarwal et al [2] and now a standard tool in the toolkit of RL theorists. Please let us know if you have further questions on this and we would be happy to address.
> > >
> > > ---
> > >
> > > **References**
> > >
> > > [1] Tong Zhang. From ε-entropy to KL-entropy: Analysis of minimum information complexity density estimation. The Annals of Statistics, 34(5):2180–2210, 2006.
> > >
> > > [2] Alekh Agarwal, Sham Kakade, Akshay Krishnamurthy, and Wen Sun. Flambe: Structural complexity and representation learning of low rank mdps. NeurIPS, 2020.

---

> > > > ### Comment · Reviewer_2cH2 · 2024-11-19
> > > >
> > > > Dear authors,
> > > >
> > > > Thank you for your clarifications. I agree that if $\mathcal{M}$ represents the space of transition models and reward models, it should naturally bound the size of the space for marginal trajectory densities.
> > > >
> > > > However, based on the "Preliminaries" section, it seems that you are allowing different transition models and reward models at each stage. If this is the case, it suggests that $\mathcal{M}$ is defined as $M := \cup_{h=1}^H (\mathbb{T}_h, \mathbb{O}_h, r_h)$. Under this assumption, $|\mathcal{M}|$ would scale with $H$. Conversely, if you are assuming that the transition and reward models remain unchanged across stages, then I agree with your conclusion.
> > > >
> > > > Reviewer 2cH2

---

> > > > > ### Author Response · Authors · 2024-11-19
> > > > >
> > > > > Thanks for the clarification! We believe we might understand your confusion (and misunderstanding) now.
> > > > >
> > > > > You are right that we do allow a **single** POMDP model $M$ to have non-stationary (also called time-inhomogeneous) transition, emission, and reward functions. But that does not lead to the conclusion that $|\mathcal{M}|$ scales with $H$. We believe you confused a single POMDP $M$ (in "POMDP setup" in the preliminary section) and the POMDP class $\mathcal{M}$ (in Assumption B, which consists of many POMDPs $M$). On a related note, your notation $\bigcup\_{h=1}^H (\mathbb{T}\_h, \mathbb{O}\_h, r\_h)$ is inaccurate, as $(\mathbb{T}\_h, \mathbb{O}\_h, r\_h)$ is a tuple, not a set, and we cannot take union $\bigcup$ over them.
> > > > >
> > > > > What you probably mean is that $\mathcal{M}$ is a "cartesian-product" class, i.e., for each stage $h$, we can choose the transition (and emission & reward) from a "base class", and transitions at different stages can be chosen independently, thus the size of the "final" model class is the size of the "base class" to the power of $H$. (On a related note, the correct notation to express this is to take the cartesian product "$\times$" across those base classes, not $\bigcup$ as you suggested.) If this is a correct interpretation of what you have in mind, then you mistakenly think $\mathcal{M}$ is that "base class", whereas in our paper $\mathcal{M}$ is the "final class", where each member is a full specification of the transition/emission/reward for all levels (i.e., $M$ in preliminaries). In fact, our setup is more general and subsumes this "cartesian-product" case, and the $H$ dependence would be implicit in $\log|\mathcal{M}|$ since $\mathcal{M}$ is that "final" class.
> > > > >
> > > > > Hope that helps and we are happy to address further questions.

---

> > > > > > ### Comment · Reviewer_2cH2 · 2024-11-19
> > > > > >
> > > > > > Thanks for the clarification. I understand that $\mathcal M$ is your "final" class in your paper, and I think it would be helpful if you discuss on the implicit H-dependence in log$|\mathcal M|$, as it appears in the main theorem.

---

> > > > > > > ### Author Response · Authors · 2024-11-19
> > > > > > >
> > > > > > > Thanks. We will add discussion and clarification.

---

### Official Review · Reviewer_TkqZ · 2024-11-05

**Soundness:** 2
**Presentation:** 3
**Contribution:** 2
**Rating:** 6
**Confidence:** 2

**Summary:**

This paper explores the off-policy evaluation (OPE) of history-dependent policies within partially observable Markov decision processes (POMDPs) using large observation spaces. Traditional methods for OPE in POMDPs have relied on model-free approaches and simplified memoryless policies due to computational complexity. This paper address this by examining the statistical challenges of evaluating more complex, history-dependent policies and by proposing a model-based method to circumvent limitations in model-free approaches. Key theoretical contributions include identifying scenarios where model-free methods are insufficient, as well as demonstrating how a simple model-based approach can achieve polynomial sample complexity.

**Strengths:**

It seems interesting to me that the model-based methods, which use synthetic data from learned models, bypass the limitations of model-free algorithms, especially in history-dependent scenarios.

**Weaknesses:**

1. What is the practical meaning of assumption D (multi-step outcome revealing) and how is it related to other revealing conditions in offline and online partially observable reinforcement learning, as well as the corresponding results?

2. Although it is said that $M^*$ does not necessarily satisfy the realizability, would the observable-equivalent realizability actually similar to it, and the resulting dependency on the realizability error seems actually trivial? Since the only difference is the additional gap due to the error in estimation likelihood from the misspecification of $M$.

**Questions:**

See weaknesses

---

> ### Author Response · Authors · 2024-11-15
>
> We thank the reviewer for their valuable comments and respond to the weaknesses below.
>
> ---
>
> **Assumption D**
>
> As mentioned on Line 195, multi-step outcome revealing essentially means that the future $f\_h$ can nontrivially predict the latent state $s\_h$. In particular, $C\_{\mathcal{F}} < \infty$ is equivalent to the outcome matrix $U\_{\mathcal{F},h}$ is full-rank. Roughly speaking, this means that starting from different latent states, the distribution of future observables is different under the behavior policy.
>
> The more popular revealing condition is single-step outcome revealing (Assumption E) and its subtle variants, which come from the online POMDP literature, e.g., Liu et al. (2022a) that we cited. Since this condition requires that *the immediate observation $o\_h$ nontrivially predicts $s\_h$ (Line 249)*, it is (generally speaking) stronger than multi-step outcome revealing, as the latter allows using the entire future---which includes $o\_h$ as part of it---to predict $s\_h$. In fact, one of contributions is exactly to study whether model-free and model-based algorithms can achieve polynomial guarantees under these assumptions (see columns of Table 1). For offline RL literature, the line of work by Uehara et al. (2023a) and Zhang & Jiang (2024) are the only one that we know who provide guarantees based on belief coverage (Assumption C) instead of exponential quantities (such as the boundedness of cumulative importance weight; see discussion in the introduction of Zhang & Jiang (2024)), and their assumption is essentially a version of multi-step outcome coverage.
>
> ---
>
> **Observable-equivalent Realizability**
>
> The reviewer wrote “would the observable-equivalent realizability actually [be] similar to it”. We guess you mean that the result and analysis in Theorem 8 is not technically that much different from the earlier analyses based on realizability, and incorporating the misspecification error is technically mundane. We agree, but **incorporating misspecification is not the main message of this section**. The main message is to consider state space mismatch, a practical scenario that is often overlooked by existing theoretical works on learning POMDPs. In particular, Theorem 7 may strike as a surprise since it is a natural extension of Theorem 4 to the state mismatch setting if one does not carefully think about what the assumptions really mean and which models they are really imposed on ($M^*$ vs. models in $\mathcal{M}$). Theorem 8 completes the message of Theorem 7 by showing that, polynomial guarantees can be recovered when we impose Assumption D on the models in $\mathcal{M}$ instead of $M^*$. Handling misspecification is just a nice little bonus orthogonal to the main point of the section, and you are right in pointing out that it is not technically significant.

---

> > ### Comment · Reviewer_TkqZ · 2024-11-27
> >
> > I've read the response. Thank you for the clarification.

---

### Meta-Review · Area_Chair_FUim · 2024-12-21

**Metareview:**

This paper explores off-policy evaluation (OPE) of history-dependent policies in POMDPs with large observation spaces, establishing theoretical hardness results for model-free approaches while proposing a model-based alternative that achieves polynomial sample complexity. The work provides valuable theoretical insights and clear analysis, though focusing primarily on statistical rather than computational efficiency.

**Additional Comments On Reviewer Discussion:**

During the discussion, reviewers raised several key concerns: (1) the computational inefficiency of the pre-filtering and MLE steps, (2) potential horizon-dependence of model class complexity, and (3) lack of empirical validation. The authors clarified that their focus is on statistical tractability rather than computational efficiency, explained that the model class does not explicitly scale with horizon length, and acknowledged the practical challenges of implementing MLE in latent-variable models. The theoretical contributions and clear analysis addressing fundamental questions about model-free versus model-based approaches in this setting, combined with the thorough handling of state-space mismatch scenarios, make this paper worthy of acceptance despite its primarily theoretical nature.

---

### Decision · Program_Chairs · 2025-01-22

Accept (Poster)